Citation: *Molecular Systems Biology* 9:696
www.molecularsystemsbiology.com

# A negative genetic interaction map in isogenic cancer cell lines reveals cancer cell vulnerabilities

Franco J Vizeacoumar[1,2,13], Roland Arnold[1,13], Frederick S Vizeacoumar[3], Megha Chandrashekhar[1], Alla Buzina[1], Jordan TF Young[3,4], Julian HM Kwan[1,4], Azin Sayad[1], Patricia Mero[1], Steffen Lawo[3,4], Hiromasa Tanaka[1], Kevin R Brown[1], Anastasia Baryshnikova[1,4], Anthony B Mak[1], Yaroslav Fedyshyn[1], Yadong Wang[5], Glauber C Brito[1], Dahlia Kasimer[1], Taras Makhnevych[1], Troy Ketela[1], Alessandro Datti[3], Mohan Babu[6], Andrew Emili[1,4], Laurence Pelletier[3,4], Jeff Wrana[3,4], Zev Wainberg[7], Philip M Kim[1,4,8], Robert Rottapel[5,9,10], Catherine A O'Brien[5,11,12], Brenda Andrews[1,4], Charles Boone[1,4] and Jason Moffat[1,4,*]

[1] Donnelly Centre and Banting and Best Department of Medical Research, University of Toronto, Toronto, Ontario, Canada, [2] Saskatchewan Cancer Agency, Department of Biochemistry, University of Saskatchewan, Saskatoon, Saskatchewan, Canada, [3] Samuel Lunenfeld Research Institute, Mount Sinai Hospital, Toronto, Ontario, Canada, [4] Department of Molecular Genetics, University of Toronto, Toronto, Ontario, Canada, [5] Campbell Family Institute, Ontario Cancer Institute, Princess Margaret Hospital, University Health Network, Toronto, Ontario, Canada, [6] Department of Biochemistry, Research and Innovation Centre, University of Regina, Regina, Saskatchewan, Canada, [7] Jonnson Comprehensive Cancer Center, Geffen School of Medicine, University of California at Los Angeles, Los Angeles, California, USA, [8] Department of Computer Science, University of Toronto, Toronto, Ontario, Canada, [9] Department of Medical Biophysics, University of Toronto, Toronto, Ontario, Canada, [10] Division of Rheumatology, Department of Medicine, St. Michael's Hospital, Toronto, Ontario, Canada, [11] Department of Laboratory Medicine and Pathology, and Department of Surgery, University of Toronto, Toronto, Ontario, Canada and [12] Department of Surgery, University Health Network, Toronto, Ontario, Canada

[13] These authors contributed equally to this work

* Corresponding author. Donnelly Centre and Banting and Best Department of Medical Research, University of Toronto, 160 College Street, Toronto, Ontario, Canada M5S 3E1. Tel.: + 1 416 978 0336; Fax: + 1 416 946 8253; E-mail: j.moffat@utoronto.ca

**Improved efforts are necessary to define the functional product of cancer mutations currently being revealed through large-scale sequencing efforts. Using genome-scale pooled shRNA screening technology, we mapped negative genetic interactions across a set of isogenic cancer cell lines and confirmed hundreds of these interactions in orthogonal co-culture competition assays to generate a high-confidence genetic interaction network of differentially essential or differential essentiality (DiE) genes. The network uncovered examples of conserved genetic interactions, densely connected functional modules derived from comparative genomics with model systems data, functions for uncharacterized genes in the human genome and targetable vulnerabilities. Finally, we demonstrate a general applicability of DiE gene signatures in determining genetic dependencies of other non-isogenic cancer cell lines. For example, the $PTEN^{-/-}$ DiE genes reveal a signature that can preferentially classify *PTEN*-dependent genotypes across a series of non-isogenic cell lines derived from the breast, pancreas and ovarian cancers. Our reference network suggests that many cancer vulnerabilities remain to be discovered through systematic derivation of a network of differentially essential genes in an isogenic cancer cell model.**

*Molecular Systems Biology* **9**: 696; published online 8 October 2013; doi:10.1038/msb.2013.54
*Subject Categories:* functional genomics; molecular biology of disease
*Keywords:* genetic interaction; genome stability; mitotic stress; pooled shRNA screening

## Introduction

A general principle in modern cancer genetics is that the analysis of genetic lesions in cancer cells should provide mechanistic insight into cancer cell biology and provide new avenues for therapeutic intervention (Bell *et al*, 2011). It is clear that cancer cells carry many mutations that are not present in their normal counterparts (Stephens *et al*, 2009; Bignell *et al*, 2010; Hudson *et al*, 2010; Greenman *et al*, 2012) and these cancer-specific mutations may represent genetic 'vulnerabilities' for tailored cancer therapy. In particular, the large-scale identification of mutations leading to genetic interactions that cause differential essentiality, contextual

lethality or synthetic sickness/lethal (SSL), in a cancer cell-specific genetic background, should prove particularly fruitful (Hartwell *et al*, 1997). For example, mutation of *BRCA1* or *BRCA2*, paralogous genes that control DNA repair, is associated with breast cancer and causes cell death when *PARP1* is also mutated (Bryant *et al*, 2005; Farmer *et al*, 2005). *PARP1* encodes for poly (ADP-ribose) polymerase (Bryant *et al*, 2005; Farmer *et al*, 2005) and inhibition of *PARP1* in *BRCA* mutant cells results in the persistence of DNA damage leading to lethality (Bryant *et al*, 2005; Farmer *et al*, 2005). Importantly, DNA damage is only one of the stress phenotypes of cancer cells that can be exploited through synthetic lethal approaches

to reveal therapeutically relevant genetic interactions (Luo *et al*, 2009b).

The largest efforts to map genetic interactions have been in model systems, principally the budding yeast, and these experiments have shown that genetic interaction networks are rich in functional information, enabling the discovery of new biological pathways and prediction of gene function (Lehner *et al*, 2006; Costanzo *et al*, 2010; Horn *et al*, 2011). Recently, model organism genetic-interaction maps have been used to direct experiments in cancer cells. For example, a cross-species synthetic lethal candidate gene approach correctly predicted a conserved synthetic lethal interaction between *RAD54B* and *FEN1* (McManus *et al*, 2009). However, this approach has been met with very limited success over the years (Hartwell *et al*, 1997). Nevertheless, genetically tractable model systems have been indispensable at revealing fundamental biological principles for over a century and have set the stage for constructing large-scale maps of genetic interactions in human cancer cells. Given that the conservation of genetic interactions in core biological processes (e.g., DNA replication, DNA damage response, chromatin remodeling and intracellular transport) is estimated to be ∼29% for distantly related species of yeast (Dixon *et al*, 2008), it is clear that to understand the interplay between genetic pathways in human cancer cells we must build a genetic interaction network from first principles in a model human cancer cell line. Moreover, the importance of systematically identifying genetic interactions in cancer cells is amplified by recent evidence, suggesting that genetic interactions create phantom heritability and may, in part, be at the root of missing heritability of common traits (Zuk *et al*, 2012).

Genome-wide mapping of genetic interactions in human cancer cells has become possible with the development of large-scale RNA interference (RNAi) libraries and focused efforts have been made to systematically identify negative genetic interactions in paired isogenic cancer cell lines, for example, with mutant *RAS* (Luo *et al*, 2009a) and loss of *TP53* (Krastev *et al*, 2011). An alternative screening strategy has been to use RNAi screens to identify genes required for proliferation across a panel of cancer cell lines and infer contextual lethality based on classification of the cell lines according to specific genomic features (Barbie *et al*, 2009) or cancer subtypes (Aarts *et al*, 2012). Large-scale efforts to identify differentially essential genes across cancer cell lines have shown that functional genomic and genomic classification schemes yield only partially overlapping results, implying that functional genomic studies reveal nuances in cancer cell biology that are not captured by genomic analyses alone (Cheung *et al*, 2011; Marcotte *et al*, 2012; Nijhawan *et al*, 2012; Rosenbluh *et al*, 2012).

The systematic identification of genetic interactions in cancer cells holds great promise for future development of effective combination therapies for different types of cancer, but it also represents a huge logistical hurdle to accomplish (Bernards, 2012). The ultimate goal of developing a universal genetic interaction network is to define genetic dependencies of cancer cells and this requires a standardized approach that will serve to build a reference network of digenic interactions in a common genetic background. In order to advance this goal, we used an established genetic screening platform (Marcotte *et al*, 2012) to identify negative genetic interactions across a small set of isogenic human cell lines. We focused on negative genetic interactions, because these are more likely to represent putative 'targets' or yield 'drivers' for specific cancer genotypes. Strikingly, even within this small set of queries we discovered and validated hundreds of negative genetic interactions, revealed novel functional relationships for uncharacterized genes and reconfirmed some genetic principles derived from studies using model organisms.

## Results and discussion

### Identification of genetic interactions in isogenic cancer cell lines

To explore the possibility of developing a network of negative genetic interactions (i.e. SSL) in human cells, we chose six isogenic cell lines and screened these in parallel using a standardized genome-scale pooled shRNA screening pipeline previously established in our lab for identifying genes that are more essential for proliferation in breast, pancreatic and ovarian cancer cells (Figure 1A; Marcotte *et al*, 2012). The HCT116 genetic background was chosen because it is near diploid with intact DNA damage and spindle checkpoints (Waldman *et al*, 1996), HCT116 cells are genetically tractable with gene replacement technologies (Waldman *et al*, 1996) and there is a large number of derived cell lines that are well characterized and available for study (Shirasawa *et al*, 1993; Jallepalli *et al*, 2001; Traverso *et al*, 2003; Lee *et al*, 2004; Hiyama *et al*, 2006). The 'query' genotypes chosen were $PTTG1^{-/-}$, $BLM^{-/-}$, $MUS81^{-/-}$, $PTEN^{-/-}$ and $KRAS^{+/-}$ (Figures 1B–F). These queries represent a functionally diverse set of genes involved in different biological processes. We screened the parental cell line and each of the five query or 'mutant' cell lines in biological triplicate using a pool of 78 432 unique shRNAs targeting 16 056 human genes (Moffat *et al*, 2006; Marcotte *et al*, 2012), testing ∼400 000 gene–shRNA or ∼80 000 gene–gene interactions. For each replicate screen, we examined multiple time points as the populations proliferated and evolved in culture, and observed very good correlation between replicates (Supplementary Figures S1A and B; $R = 0.9$–0.99 for replicates). The abundance trend of each hairpin at different timepoints was used to compute a dynamic hairpin-

**Figure 1** (**A**) Workflow for the identification of genetic interactions using pooled shRNA screens. E, essential gene; NGI, negative genetic interaction. (**B–F**) Simplified schematics showing the major functions of the five query genes (boxed in bold) that were screened for negative genetic interactions. (**G–K**) Scatter plot of zGARP scores for parental and query cell lines from primary screens. Negative genetic interactions ($P<0.05$) with the query genes are indicated in red, whereas genes that are essential in both parental and mutant lines are highlighted in blue. (**L**) Experimental setup for the HCS-based co-culture assay and representative images from *PTTG1* competition assay at days 2 and 7 for control and DHFR knockdown conditions. The $PTTG1^{+/+}$ cells are in red and the $PTTG1^{-/-}$ cells are in green. The green and red vertical bars at the side of each image represent the proportion of red/green cells within that image as determined by image analysis. (**M–Q**) Results from the secondary competition assays for each of the five query genes. The *y*-axis represents the average fitness ratio of mutant cells versus parental cells ($n = 2$). The black line represents the mean of the lowest drop from mock-transfected cells ($n = 25$). The green dotted lines represent the highest and lowest distribution of the mock-transfected cells with 80% CI. Red dots are genes that were examined in more detail in tertiary assays.

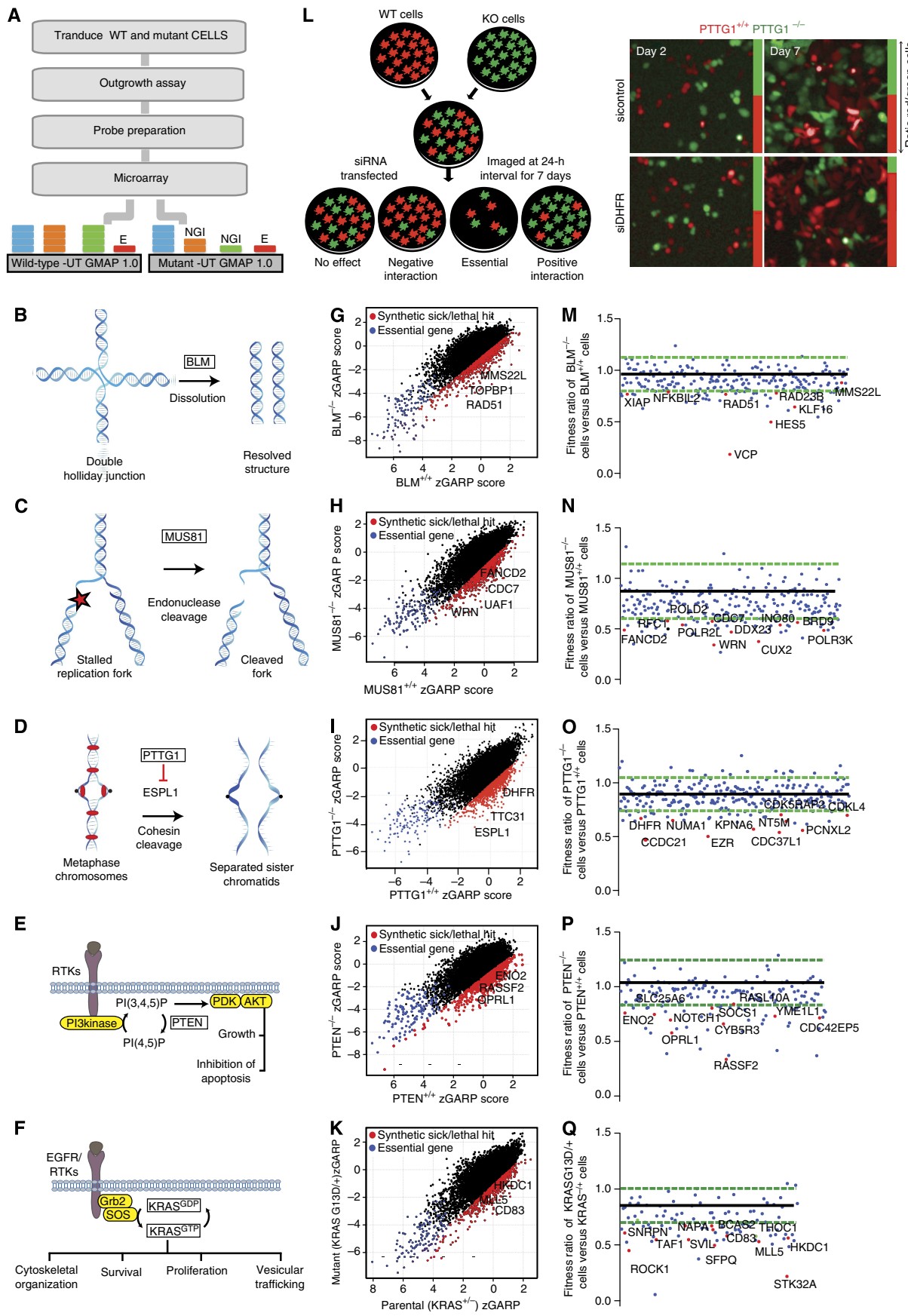

level score termed shARP (Supplementary Figures S1G–K and Supplementary Table S1) and, from this, a gene-level essentiality score termed GARP, which is the average shARP scores for the top two performing shRNAs for each gene (Supplementary Information; Marcotte *et al*, 2012). A scatter plot of the normalized GARP (i.e., zGARP) scores for each query gene compared with the parental control cell line revealed many candidate genetic interactions by differential zGARP (i.e., dGARP) for each of the query genes (Figures 1G–K). To reduce the number of false positives inherent to RNAi screens, we also performed genome-scale microarray gene expression profiling experiments on parental HCT116 cells and in all five query cell lines in order to measure target mRNA levels, and used these levels to determine a threshold for the presence/absence (see Supplementary Information). Therefore, using stringent negative dGARP scores ($P < 0.05$; Supplementary Figures S1L–P and Supplementary Table S2) and filtering for mRNA target gene expression (i.e., the presence/absence; see Supplementary Table S3), we generated a network of negative genetic interactions across the five query genes consisting of 2014 nodes and 2617 edges, which we will refer to hereafter as a differential essentiality (DiE) network (Supplementary Figure S2A).

In order to confirm a subset of the negative genetic interactions identified in our DiE network for each of the query genotypes, and to help rule out that our primary screening results were not due to RNAi off-target effects (Moffat *et al*, 2007; Kaelin, 2012), we designed five separate colored competition assays, one for each query, where equal numbers of parental cells expressing red fluorescent protein and query cells expressing green fluorescent protein were mixed and cocultured following knockdown of putative SSL genes using orthogonal siRNA reagents (Figure 1L and Supplementary Information). Putative SSL genes were selected for confirmation by considering target expression, GARP scores ($P < 0.05$), shARP scores ($P < 0.01$), differential gene expression between the parental and query cells, and yeast orthology (Supplementary Figure S2B), as we anticipated exploiting the large-scale mapping of genetic and chemical–genetic networks from yeast to validate some of the genetic interactions we observe in the DiE network (Supplementary Figure S2A). Mixtures of parental and query cells were reverse transfected with siRNAs and the relative ratio of red-to-green cells, respectively, was determined every day for 7 days by automated microscopy and image analysis (Figure 1L) after normalizing for growth differences between the parental and derived cell lines (Supplementary Figures S2C–G). As siRNAs can have a transient effect over the course of 7 days, we used the greatest fold-change in red-to-green cells from two independent replicates to calculate a relative fitness score for query versus parental cells. A total of 826 genetic interactions were tested across five assays and 200 unique negative genetic interactions (24.2%) were confirmed to differentially decrease the fitness of the query cells compared with the parental cells (80% CI; Figure 1M–Q and Supplementary Tables S4 and S5).

We tested a small number of interactions across all the five colored competition assays to examine the specificity of each genetic interaction. For example, *HKDC1* was identified in the *KRAS* primary screen and knockdown in the secondary assays showed synthetic lethality specific to the *KRAS*$^{G13D/-}$

genotype (Supplementary Figure S2H). Another example is *KPNA6*, which was identified in both the *PTTG1* and *MUS81* primary screens, and was subsequently shown to be synthetic lethal only in these genotypes in the secondary validation screens (Supplementary Figure S2I). In contrast, some of the hits from the primary screen were not validated in the secondary assays (e.g., *ESPL1* in *PTTG1* assay; Supplementary Figure S2J), but were validated using shRNAs (see below). Lastly, hits like *RASSF2*, which was identified in the primary *PTEN* screen, also validated in the secondary screen (Supplementary Figure S2K). However, *RASSF2* knockdown was also found to be synthetic lethal with *KRAS*$^{G13D/-}$ in the secondary screen (Supplementary Figure S2K), but did not emerge as a hit in the primary *KRAS* screen. This small set of specificity tests demonstrates that our secondary validation assays largely recapitulate the specificity of our primary genetic interaction data.

As mentioned above, one of our goals was to use comparative genomics to identify genetic interactions that may be conserved between HCT116 cells and model systems. By using InParanoid to identify orthologs, which depends on bidirectional best BLAST hits (O'Brien *et al*, 2005), we observed that several established functional orthologs were not represented. For example, human *ESPL1* shows a strong negative genetic interaction with *PTTG1* and is orthologous to yeast *ESP1* (Ciosk *et al*, 1998), which was not reported in InParanoid. Thus, using overly stringent orthology mapping will underestimate the number of conserved genetic interactions. To identify a larger set of functional orthologs, we developed a less stringent orthology mapping algorithm to identify the most probable orthologs of human genes in model organisms based on eggNOG 2.0 (Muller *et al*, 2010), which we refer to as MP-eggNOG (Supplementary Figure S3A and Supplementary Table S6). The eggNOG uses orthologous clusters to detect protein relationships, resulting in an increased number of cross-species orthologs compared with alternative algorithms (Supplementary Figures S3B and C), whereas the MP-eggNOG introduces an additional filtering step that prevents overestimation of orthologous relationships (Supplementary Figure S3D and see Supplementary Information). Overall, 200 negative genetic interactions were confirmed from the five query screens ($P < 0.05$; Supplementary Table S5) and 90 non-redundant interactions were predicted as conserved (Supplementary Table S7). A high-confidence DiE network of confirmed and/or conserved negative genetic interactions containing 264 nodes and 291 edges is shown in Figure 2.

## Properties of the differential essentiality network

Our high-confidence DiE network (Figure 2) in cancer cells recapitulates some general properties of model organism genetic networks. For example, these networks are highly complex, with query genes showing dozens of confirmed interactions on average (Figure 2). We next considered how the DiE network of negative genetic interactions could be used for: (a) assessing functional conservation, (b) building functional modules, (c) discovering functions for uncharacterized genes, (d) revealing targetable vulnerabilities and (e)

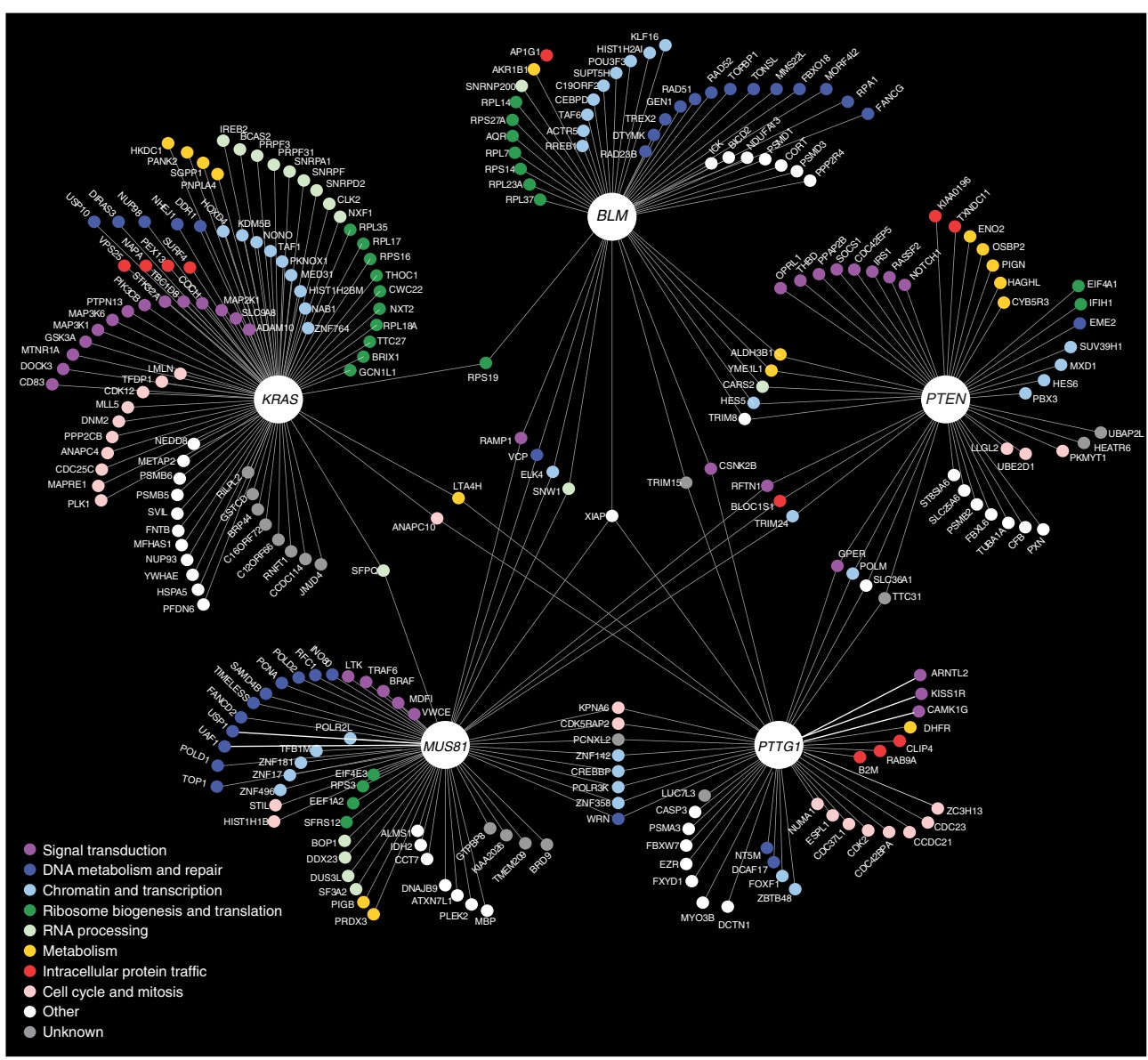

**Figure 2** Differential essentiality or DiE map. High-confidence network of confirmed and/or conserved negative genetic interactions represented as edges connected to one or more of the five query genotypes described in the text, including *KRAS*, *PTTG1*, *PTEN*, *MUS81* and *BLM*. Genes are represented as nodes and these are color coded according to the biological functions indicated in the legend.

evaluating the relationship between differential essentiality and differential mRNA expression.

## Functional conservation

One objective of the genetic interaction mapping in model systems such as *S. cerevisiae* is to provide a template for predicting conserved interactions in mammalian cells. Using InParanoid and MP-eggNOG to map orthologs, we did not find significant conservation of genetic interactions between our data and the yeast genetic interaction network (Supplementary Table S7), which may reflect the sparse overlap between genes tested in both networks. For example, genetic networks in yeast are currently underrepresented for highly conserved essential genes (Li *et al*, 2011). Nonetheless, we identified several biologically compelling examples of conserved genetic

interactions and predicted 65 non-redundant conserved interactions in different model systems across all five of our queries (Supplementary Figure S2A and Supplementary Table S7). For example, several interactions we identified for *KRAS* were supported by data from other species including: *P91029*, *UNC62* and *Q8l120* in worms and *PP2A*, *MPIP* and *TFDP* in fly cells (Supplementary Figure S3E and Supplementary Table S7).

One of the queries in our screens, *PTTG1*, encodes the evolutionarily conserved protein securin, which inhibits separase (encoded by *ESPL1*), a protein that is important for progression through mitosis (Figure 1D and Zou *et al*, 1999). As mentioned above, we observed a strong negative genetic interaction between *PTTG1* and *ESPL1*/separase ($P < 0.001$; Figure 1I and Supplementary Figure S4A). Separase is a protease that cleaves cohesin to promote sister chromatid

separation during mitosis, and loss of securin or separase leads to genome instability and a delay in anaphase progression in HCT116 colon cells (Supplementary Figure S4B and Jallepalli *et al*, 2001) and in model systems. It is noteworthy that securin and separase have a negative genetic interaction in budding yeast (Ciosk *et al*, 1998), suggesting that the functional connection between the securin and separase genes is evolutionarily conserved.

We also recovered a conserved negative genetic interaction with *BLM* and *TOPBP1* (Figure 1G). Deletion of the orthologous genes, *SGS1* and *DPB11*, in *S. cerevisiae* results in synthetic lethality manifested as a synergistic increase in the gross chromosomal rearrangement rate, a characteristic feature of genomic instability, and DNA replication damage (Myung and Kolodner, 2002). In mammalian cells, *TOPBP1* activates *ATR*, a major regulator of the DNA damage response that is present at sites of replication damage, which can be assayed by measuring γ-H2AX foci, a reporter of DNA damage that marks DNA double-strand breaks and stalled replication forks (Paulsen *et al*, 2009). Accordingly, we examined whether the absence of both *BLM* and *TOPBP1* causes accumulation of unrepaired DNA breaks by monitoring γ-H2AX foci in $BLM^{-/-}$ cells knocked down for *TOPBP1* and found that the *BLM-TOPBP1* double-mutant cells displayed 3.5-fold more γ-H2AX foci (Supplementary Figures S5A–C), suggesting that loss of both *BLM* and

*TOPBP1* compromises *ATR* signaling in response to spontaneous DNA damage, resulting in stalled DNA replication forks, enhanced genome instability and subsequent lethality (Supplementary Figure S5D).

## Functional modules

One possible outcome of the DiE network is that the genes that are connected to each of the query genes are more functionally related to each other than a random set of expressed genes. In order to investigate the relationship between genes in the DiE network, we gathered evidence of genetic or physical interaction data in the literature, including data from model organism studies, then propagated subnetworks for each query. Clusters of densely connected functional modules emerged for each of the five query genes (Supplementary Figures S6A–E and Supplementary Table S8). For example, one functional module with the query gene *MUS81* revealed a densely connected network of genetic and physical relationships (Figure 3A), including negative genetic interactions with different members of the same protein complex, including *FANCD2*, *UAF1* and *USP1* (Cohn *et al*, 2007; Murai *et al*, 2011).

*MUS81* encodes an endonuclease that cleaves 3′-ends at stalled replication forks to help repair DNA interstrand crosslinks (ICLs) along with other nucleases (Wang *et al*, 2011), and *FANCD2* encodes a key regulator of the Fanconi Anemia (FA) pathway that

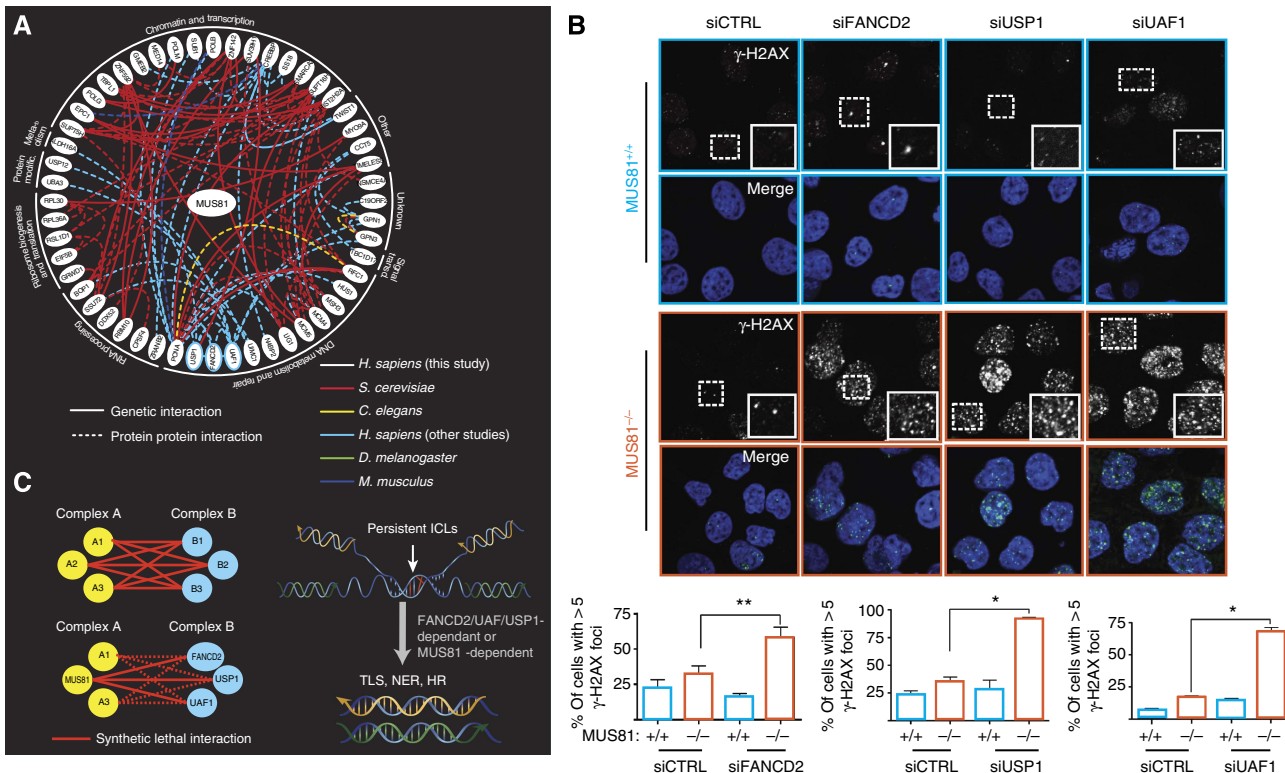

**Figure 3** Propagation of *MUS81* interaction module using comparative genetics. (**A**) A subnetwork of genetic and physical interactions between genes identified as negative interactions with $MUS81^{-/-}$ propagated after comparative cross-species analysis. Solid lines represent genetic interactions and dotted lines represent physical interactions. The color of the line indicates which model system the data was derived from (see Supplementary Information for details). (**B**) HCT116 cells, either $MUS81^{+/+}$ or $MUS81^{-/-}$ as indicated, were transfected with siRNA against *UAF1*, *USP1* or *FANCD2*, and stained using an antibody against γ-H2AX after 48 h of knockdown. The corresponding bottom panels show the percentage of cells ($n>100$) with $>5$ foci ($n=3$). **$P<0.01$ and *$P=0.02$ using Student's *t*-test. Scale bar, 12 μm. (**C**) The left side depicts a model of 'between-complex' genetic interactions and how *MUS81* and *FANCD2-UAF1-USP1* fit into this model. This model preserves genomic integrity in the face of persistent interstrand crosslinks or ICLs by providing a genetic buffering mechanism, depicted on the right.

also repairs ICLs. ICLs present a threat to genome stability, as they inhibit essential processes such as DNA replication and transcription (Cohn *et al*, 2007). Chromatin-associated *FANCD2* must be de-ubiquitinated in order to maintain a non-chromatin-bound pool of *FANCD2* to repair ICLs throughout the genome, and UAF1 and USP1 physically interact in a complex with FANCD2 to promote de-ubiquitination of FANCD2 and maintenance of the non-chromatin-bound pool of FANCD2 (Cohn *et al*, 2007). Consistent with the idea that *MUS81* functions in a pathway parallel to the FA pathway to repair persistent ICLs along with other nucleases (Wang *et al*, 2011), we observed that $MUS81^{-/-}$ cells depleted for *UAF1* contained 3.5-fold more $\gamma$-H2AX foci than isogenic $MUS81^{+/+}$ cells (Figure 3B). We also observed a significant increase in $\gamma$-H2AX foci in $MUS81^{-/-}$ cells depleted for either USP1 or FANCD2 compared with $MUS81^{+/+}$ control cells (Figure 3B). Taken together, our results indicate that *UAF1*, *USP1* and *FANCD2* are critical in the absence of *MUS81* for the repair of damaged DNA and help delineate a parallel genetic pathway involved in repair of ICLs (Figure 3C). To date, it has been difficult to compare genetic interaction data across species and between different studies; however, the idea of combining genetic and physical interaction data from studies in model systems with different orthology mapping tools to predict 'functional modules' can lead to dense clusters of genes that are functionally related in human cells.

## Functional discovery

As observed for the genetic interaction landscape of *S. cerevisiae*, the position and connectivity of genes on the genetic interaction network provides a precise prediction of gene function, resolving pathways and identifying their regulators and functional connections (Costanzo *et al*, 2010). Approximately 23% of the genes in our high-confidence DiE network had uncharacterized functions (see Supplementary Information). For example, *TTC31* is an uncharacterized gene that encodes a tetratricopeptide repeat-containing protein that was found to be more essential for proliferation in $PTTG1^{-/-}$ and $PTEN^{-/-}$ cells compared with their $PTTG1^{+/+}$ and $PTEN^{+/+}$ counterparts, respectively (Figure 2). We validated these observations by rescuing the synthetic lethal effect in $PTTG1^{-/-}$ cells following knockdown with sh2-*TTC31* by expressing a *TTC31* shRNA-resistant construct. Briefly, we constructed wild-type (*TTC31*-V5) and shRNA-resistant (*TTC31*-shR-V5) versions of TTC31 that were epitope tagged with V5 and found that the expression of TTC31-shR-V5 allowed for proliferation in $PTTG1^{-/-}$ cells in the presence of sh2-*TTC31*, whereas *TTC31*-V5 did not allow for proliferation in the presence of sh2-*TTC31* expression (Figures 4A and B).

On the basis of its genetic interaction with *PTTG1*, we hypothesized that *TTC31* functions in maintaining chromosome stability, and found that knockdown of TTC31 with two independent shRNAs in HCT116 cells resulted in bipolar spindles that contained multiple *NEDD1*/Pericentrin structures, which are considered structural markers of the centrosome (Figures 4C and D). Notably, cells depleted for *TTC31* displayed chromosome congression defects (Figures 4C and E), which were validated by immunostaining for Aurora B, a kinase that regulates bi-orientation of the mitotic spindle (Supplementary Figure S7A). Knockdown of *TTC31* in

$PTTG1^{-/-}$ cells resulted in lethality with most of the cells having high heterogeneity in nuclear morphology and multiple centrosomes (Supplementary Figures S7B and C). We observed a similar phenotype of supernumerary centrosomes in HeLa cells depleted for *TTC31* (Supplementary Figure S7D), suggesting that *TTC31* regulates centrosome duplication.

To further explore the function of *TTC31* in centrosome duplication, we performed time-lapse microscopy in HeLa cells stably expressing *NEDD1*-GFP (Lawo *et al*, 2009) in the presence and absence of *TTC31* following knockdown of TTC31 with two independent shRNAs. Formation of additional centrosome structures was observed in ~15% of mitotic cells (Figure 4F). We hypothesized that these additional centrosome structures may be due to either overduplication of centrioles or defects in centrosome integrity. To test this idea, we monitored centrioles in HCT116 cells and found that *TTC31*-depleted cells in mitosis frequently had more than four centrioles within a single cell (Figure 4G). Taken together, these results suggest that overduplication of centrioles are the cause of multiple centrosomes in *TTC31*-depleted cells. Moreover, the negative genetic interaction between *TTC31* and *PTTG1* likely reflects both the failure of *TTC31*-depleted cells to properly regulate centrosome duplication and chromosome instability in $PTTG1^{-/-}$ cells, the combination of which results in mitotic catastrophe.

An additional example of a gene with a strong negative genetic interaction with $PTTG1^{-/-}$ was *ZC3H13* ($P<0.05$), a previously uncharacterized gene exhibiting frame-shift mutations in gastric and colorectal cancers with microsatellite instability (Wang *et al*, 2004). We confirmed this synthetic sick/lethal effect using two independent shRNAs targeting *ZC3H13*, each of which were capable of knocking down the 75-kDa isoform of ZC3H13; sh2-*ZC3H13* also knocked down the 240-kDa isoform of ZC3H13 (Figure 4H). Importantly, we also noticed that the levels of two ZC3H13 isoforms fluctuated across multiple pancreatic cancer lines and that a portion of the ZC3H13 protein shared sequence homology with the yeast PDS1/securin protein (Supplementary Figures S8A and B). ZC3H13 is regulated in a cell cycle-dependant manner, with levels peaking in the G1–S phase of the cell cycle and degradation occurring before the G2–M phase and accumulation of PTTG1/securin (Figure 4I). In the absence of *PTTG1*, ZC3H13 levels persist into mitosis as indicated by the peak level of cyclin B1 (Figure 4I), suggesting that accumulation of PTTG1 correlates with destruction of ZC3H13 in cells that express PTTG1.

We next explored the effects of knocking down ZC3H13 on mitotic phenotypes using the mitotic marker phospho-histone H3 and the centrosomal marker NEDD1. In control cells when the NEDD1 markers were 6–7 mm apart, a metaphase plate was clearly evident (Figures 4J and K). By contrast, in cells knocked down for ZC3H13, where the NEDD1 markers were >6–7 $\mu$m apart, the chromosomes were largely dispersed and not congressed to the metaphase plate (Figures 4J and K). Notably, proteomic approaches identified ZC3H13 copurifying with centromeric proteins such as CENP-A (Obuse *et al*, 2004), suggesting that ZC3H13 might have a key role in chromosome segregation. Consequently, we monitored two key spindle checkpoint proteins, BUBR1 and CENP-E, in the presence of a control shRNA (shRFP), and found that in $PTTG1^{+/+}$ cells the

levels of BUBR1 and CENP-E increased at the kinetochore during prometaphase and decreased at metaphase (Figures 4L–N). By contrast, in *PTTG1*$^{+/+}$ cells depleted of ZC3H13 with two independent shRNAs (sh1- and sh2-*ZC3H13*), BUBR1 and CENP-E did not increase substantially at the kinetochores even during prometaphase (Figures 4L–N). These observations suggest that ZC3H13 has a functional role in the spindle assembly checkpoint (Supplementary Figure S8D). Taken

together, we have implicated both *TTC31* and *ZC3H13* in critical cell cycle processes by inferring these roles from genetic interactions with *PTTG1*.

### Targetable vulnerabilities
Another reason to map synthetic lethal networks in tumor cells is to identify chemical–genetic relationships that reveal possible susceptibilities. We highlight two examples of genetic

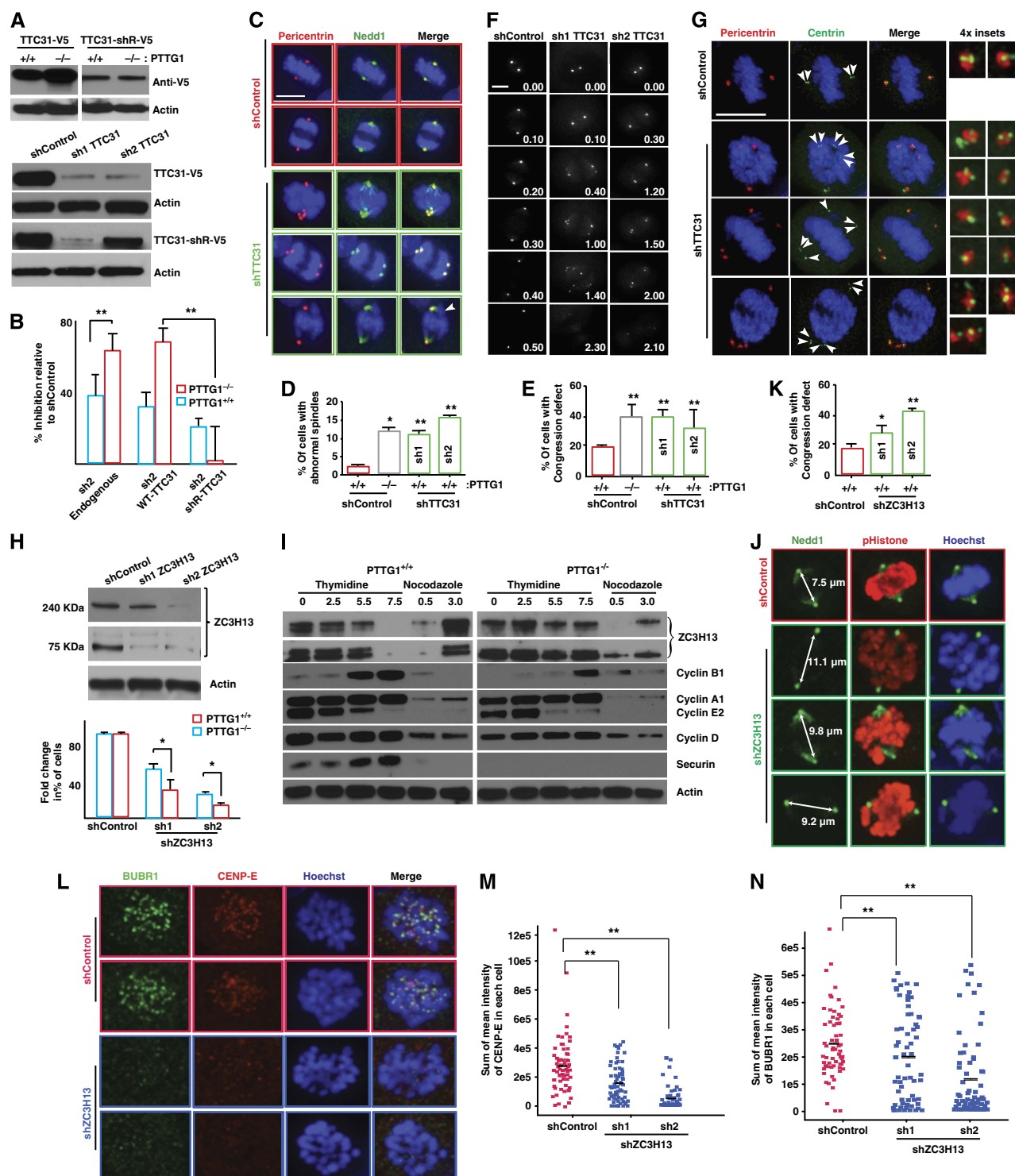

interactions in our high-confidence DiE network that represent potential cancer cell vulnerabilities. The first example is a strong negative genetic interaction between *PTTG1* and *DHFR*, an enzyme that reduces dihydrofolic acid to tetrahydrofolic acid (Figures 1I and O). It is noteworthy that this relationship was predicted in yeast based on chemical–genetic profiling experiments, which found that the yeast securin gene, *PDS1*, is haplo-insufficient when cells are treated with methotrexate (MTX), a potent and specific inhibitor of DHFR activity ($P < 1.5e - 7$; Hillenmeyer *et al*, 2008). Both *PDS1* and *DHFR* yeast genes are essential in yeast cells; hence, we first confirmed this chemical–genetic observation in yeast by treating a haploid mutant yeast strain carrying a temperature-sensitive hypomorphic allele of yeast securin, *pds1-128*, with MTX and found that these cells were indeed hypersensitive to MTX as compared with the isogenic wild-type yeast strain (Supplementary Figure S9A; $P < 0.01$). These results demonstrate that the *PTTG1–DHFR* negative genetic interaction in human cells is evolutionarily conserved and consistent with the chemical–genetic interaction in yeast between MTX and *PDS1*. On further examination of the *PTTG1–DHFR* negative genetic interaction in human HCT116 cells, we noticed that knockdown of *DHFR* with two independent shRNAs in *PTTG1$^{+/+}$* cells, or treatment of these cells with MTX, resulted in abnormal spindles and binucleated cells consistent with genomic instability (Figures 5A–C). Similar observations were also made in HeLa cells (Supplementary Figure S9B) and in HT29 cells (de Anta *et al*, 2006).

As DHFR is important for maintaining cellular nucleotide pools necessary for DNA synthesis and PTTG1 functions in cell cycle progression to prevent precocious separation of sister chromatids, we reasoned that *DHFR* and *PTTG1* might be co-expressed. By analyzing the expression data across a set of ~300 cancer cell lines, we found a positive correlation between *PTTG1* and *DHFR* expression (Supplementary Figure S9C; $R = 0.458$; $P$-value $< 0.01$). Given the clinical importance of MTX and associated resistance mechanisms

linked to cancer relapse (Schimke, 1986), we re-examined the expression profiles previously generated from seven MTX-sensitive and -resistant paired cell lines (Selga *et al*, 2009) and observed that both *PTTG1* and *ESPL1* expression were tightly correlated with sensitivity and resistance to MTX (Figure 5D; $P < 2.2 \times 10^{-16}$). These results suggest that *PTTG1* and *ESPL1* expression may be useful markers to predict heightened resistance to MTX.

The second example of a chemical–genetic relationship that represents a possible vulnerability stems from the observation of a negative genetic interaction between *KRAS$^{G13D}$* and *CD83*, which are genes that encode a cell surface protein whose function is unclear but is thought to be involved in immunosuppression (Breloer and Fleischer, 2008). Normally, wild-type KRAS is thought to act downstream of EGFR to promote proliferation, survival, motility and a number of other biological functions necessary for cellular proliferation, whereas mutant gain-of-function KRAS$^{G13D}$ circumvents many EGFR functions by acting downstream of EGFR (Figure 4E). We discovered a strong negative genetic interaction between *KRAS$^{G13D}$* and *CD83* in our primary HCT116 screens (Figure 1K) and in the secondary competition assay between *KRAS$^{G13D/+}$* and *KRAS$^{-/+}$* cells (Figure 1Q). In addition to these screens, which were performed in the HCT116 genetic background, we also screened a different colorectal cancer cell line, LIM1215 cells, which harbor wild-type alleles of *KRAS* and *EGFR*, and are sensitive to the antibody drug Cetuximab (also known as Erbitux), which blocks EGFR signaling (Figure 5E). The dropout screens in LIM1215 cells were carried out using the same genome-scale lentiviral-based shRNA-pooled approach as described above for the HCT116 query cell lines, both in the presence and absence 10 µg/ml of Cetuximab (i.e., IC$_{20}$), in order to discover genes that would sensitize LIM1215 cells to Cetuximab (Figure 5F). *CD83* showed a strong synthetic lethal effect with the addition of Cetuximab to LIM1215 cells (Figure 5G; $P < 0.05$). The synthetic lethal interaction between Cetuximab or *KRAS$^{G13D}$* and *CD83* was

---

**Figure 4** Functional discovery based on genetic interactions. (**A**) Top panels shows the western blot of total cell lysates expressing both wild-type (TTC31-V5) and shRNA-resistant (TTC31-shR-V5) TTC31 protein tagged with V5 in *PTTG1$^{-/-}$* and *PTTG1$^{+/+}$* cell lines. The bottom panel shows the effect of two independent shRNAs in TTC31-V5 and TTC31-shR-V5. TTC31 was knocked down by infecting HCT116 cells with two independent hairpins targeting *TTC31*, followed by a western blot with anti-V5 antibody to confirm on-target knockdown. (**B**) Rescue experiment showing the on-target effect of sh2-*TTC31* as evaluated by the percentage inhibition in growth WT-*TTC31*-expressing cells relative to shR-*TTC31*-expressing cells. (**C**) Immunofluorescence microscopy images of *PTTG1$^{+/+}$* cells expressing control shLacZ (top) or sh2-*TTC31* (bottom). Centrosomes are indicated by pericentrin (red) and NEDD1 (green) staining, and the nucleus was detected with DAPI (blue). Arrowheads indicate lagging chromosomes. (**D** and **E**) Quantification of cells with abnormal spindle morphology and cells with congression defects was performed by analyzing at least 100 cells ($n = 3$). **$P$-value $< 0.01$ calculated using $\chi^2$-test. Scale bar, 4 µm. (**F**) Time-lapse imaging of HeLa cells stably expressing the centrosome marker NEDD1-GFP. Cells were either infected with the negative control *sh*RNA targeting LacZ or with two independent *sh*RNAs against *TTC31* (sh1- or sh2-*TTC31*). Frames taken at the indicated time points (h:min) relative to entry into mitosis are shown. Scale bar, 10 µm. (**G**) Localization of centrin structures in *PTTG1$^{+/+}$* cells expressing shLacZ control or one of two independent *sh*RNAs against *TTC31* (sh1- or sh2-*TTC31*). DNA and centrosomes were labeled as in **C**. Representative prometaphase, metaphase and anaphase cells with supernumerary centriole phenotypes are shown. Insets are fourfold magnifications of centrosomal regions. Scale bar, 10 µm. (**H**) Top panels shows the western blot of total cell lysates expressing two isoforms of ZC3H13 in *PTTG1$^{+/+}$* cells. ZC3H13 was knocked down by infecting HCT116 cells with two independent hairpins (sh1- and sh2-*ZC3H13*) and followed by western blot with an anti-ZC3H13 antibody (Abcam). The bottom panel shows the negative genetic interaction of two independent shRNAs as assessed by proliferation. (**I**) Western blot analysis of whole-cell lysates showing the cell cycle-dependant regulation of ZC3H13 in *PTTG1$^{-/-}$* and *PTTG1$^{+/+}$* cell lines. The first four lanes represent the time after release from double thymidine block and the last two lanes from thymidine-nocodazole block. The levels of different cyclins fluctuate across different phases of cell cycle and are used as controls. (**J**) Immunofluorescence microscopy images of *PTTG1$^{+/+}$* cells expressing a negative control shRNA, shRFP (top), or sh2-*ZC3H13* (bottom). NEDD1 antibody (green) was used to stain for centrosomes and phosphohistone H3 antibody (red) to stain for mitotic cells. Nuclei were stained with Hoechst stain (blue). The distances between the poles are shown for representative cells. (**K**) Quantitation of chromosome congression defects in control as well as cells following ZC3H13 knockdown from two independent experiments. A minimum of 30 cells were counted in each case. *$P < 0.01$, Student's *t*-test. (**L**) Immunofluorescence microscopy of spindle checkpoint proteins CENP-E and BUBR1 in *PTTG1$^{+/+}$* cells expressing the control shRFP (top) or sh2-*ZC3H13* (bottom). Cells were stained with BUBR1 antibody (green) and CENP-E antibodies (red). Nuclei were stained with Hoechst stain (blue). (**M** and **N**) Quantitation of localization defect of spindle checkpoint proteins, measured as the sum of mean fluorescence intensity of BUBR1 or CENP-E in each cell. The aggregate results from two independently infected cell populations are shown with either shControl, sh1-*ZC3H13* or sh2-*ZC3H13*. Thirty to 50 cells were counted in each case. **$P < 0.001$, **$P < 0.01$ by Student's *t*-test.

---

confirmed with three independent shRNAs targeting *CD83* (sh1-, sh2- and sh3-*CD83*) in LIM1215 and HCT116-*KRAS*$^{G13D}$ cells, respectively (Figures 5H and I).

In order to further confirm these results and ensure that our observations were not due to off-target effects of the shRNAs (Moffat *et al*, 2007), we rescued the growth inhibition caused by sh1-*CD83* by expressing an shRNA-resistant CD83 cDNA construct, sh1R-CD83-GFP, in HCT116-*KRAS*$^{G13D}$ cells (Figure 5J). Both the WT-CD83-GFP and sh1R-CD83-GFP fusion proteins were found to be properly localized to the cell

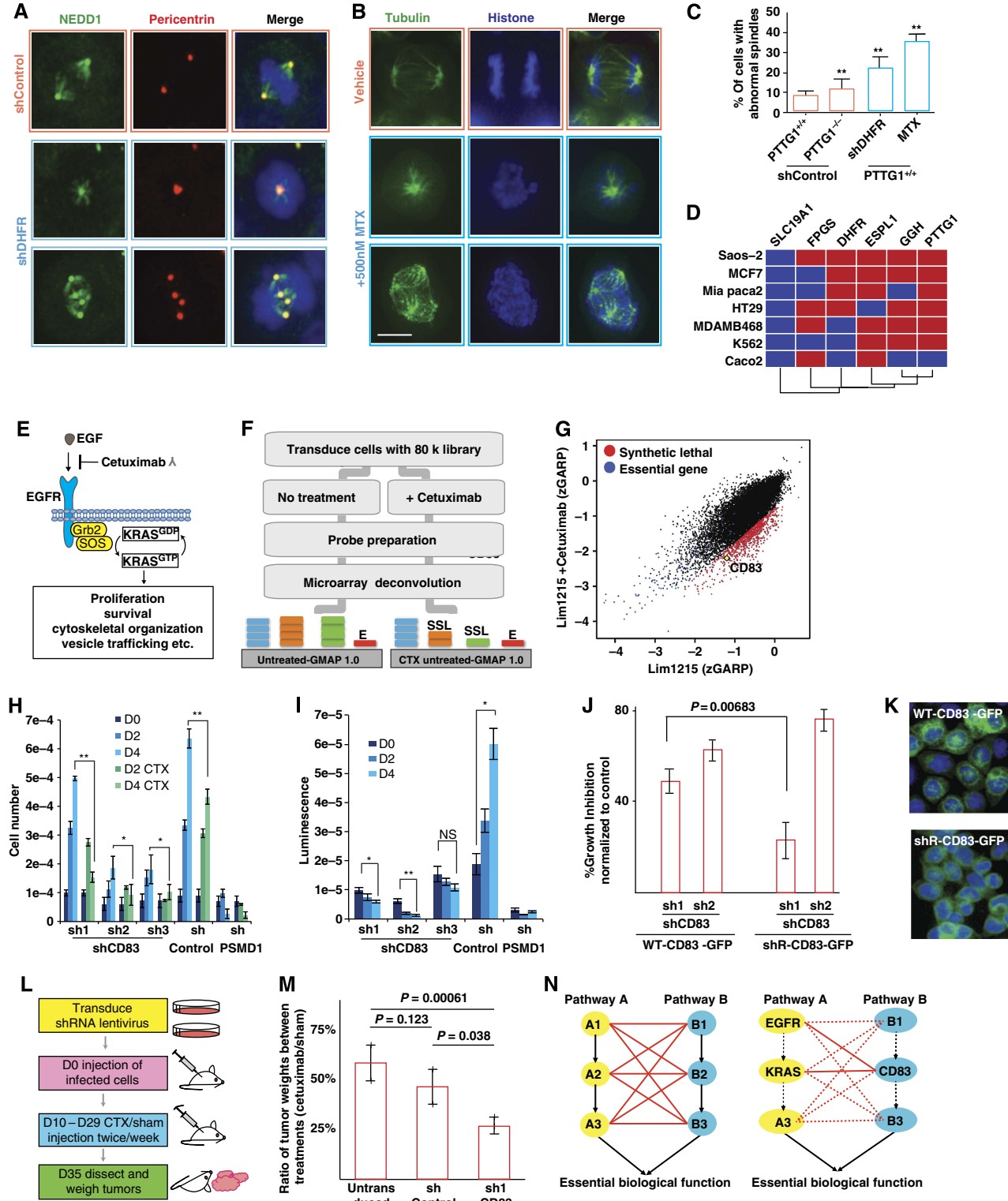

membrane in HCT116-$KRAS^{G13D}$ cells as determined by microscopy (Figure 5K). Lastly, we examined whether the genetic interaction between Cetuximab and *CD83* still occurred in the context of an *in vivo* tumor xenograft model. For this experiment, we established LIM1215 tumor xenografts in NOD/SCID mice with and without CD83 knockdown, using sh1-*CD83*, by injecting 5e6 cells subcutaneously (Figure 5L). Ten days post injection, we performed injections of Cetuximab, or a sham injection, two times weekly (Figure 5L). After 35 days, the tumors were collecetd and measured for differences in size. The sh1-CD83 knockdown tumors were > 2-fold smaller than the tumors expressing the negative control shLacZ ($P = 0.038$) or the untransduced LIM1215 cells ($P = 0.0006$; Figure 5M). Taken together, these results support the idea that *CD83* could be acting in a parallel genetic pathway with *EGFR* and *KRAS* to impinge on essential biological functions necessary for cell growth, both *in vitro* and *in vivo* (Figure 5N).

## Differential essentiality versus differential expression

Integrating differential gene expression with the DiE network adds context to certain genetic interactions (Luscombe *et al*, 2004; Ideker and Krogan, 2012). We examined the union of negative genetic interactions and differential gene expression, as increased gene expression may indicate a proliferation requirement for a specific gene product in the mutant query cells used in our screens (Figures 1B–F). Although there was no significant correlation between differential gene expression and differential essentiality amongst the query genes in our DiE network, there were some striking examples. For example, we applied this approach to the *PTEN* screen by filtering confirmed *PTEN*-negative genetic interactions with outlier genes that showed increased expression in the $PTEN^{-/-}$ mutant cells compared with the $PTEN^{+/+}$ cells. Notably, the Ras Asociation Domain Family 2 (*RASSF2*) exhibited a negative genetic interaction with *PTEN* (Figures 1J and P) and had higher expression levels of *RASSF2* in $PTEN^{-/-}$ cells relative to $PTEN^{+/+}$ cells (Figure 6A and Supplementary Figure S10A). *RASSF2* is a tumor suppressor gene that functions as an effector of *RAS* signaling (Akino *et al*, 2005). Moreover, *AKT* and *RASSF2* both regulate *MST1/2* (Cooper *et al*, 2009; Romano *et al*, 2010), a central kinase in the Hippo signaling cascade that negatively regulates Yes-associated protein 1 (*YAP1*), a transcriptional co-activator with both

oncogenic and tumor suppressor activities (Lapi *et al*, 2008; Song *et al*, 2010). We reasoned that the combined loss of *PTEN* and *RASSF2* causes cell death through the Hippo signaling cascade (Supplementary Figure S10B). Consistent with this hypothesis, we found that $PTEN^{-/-}$ cells had elevated YAP1 protein levels, which were further increased when RASSF2 was knocked down (Figure 6B and Supplementary Figure S10C). We also observed that HCT116-$PTEN^{-/-}$ cells depleted for either RASSF2 or the Hippo pathway component LATS1 showed a greater apoptotic response to cisplatin treatment (Figure 6C), consistent with the observation that *YAP1* mediates apoptosis in response to cisplatin treatment in HCT116 cells (Lapi *et al*, 2008). The dual oncogenic and tumor suppressor activities of *YAP1* provide an explanation for how the loss of two proapoptotic tumor suppressor proteins (*RASSF2* and *PTEN*) results in decreased cell survival mediated by the Hippo signaling pathway (Supplementary Figure S10B).

It is well known that the *PTEN/AKT* pathway promotes glucose uptake and metabolic reprogramming in cancer cells (Elstrom *et al*, 2004). Our data revealed a strong genetic interaction between *PTEN* and the glycolytic enzyme *ENO2* (Figure 1J) that was confirmed in a *PTEN* competition assay (Figure 1P). Remarkably, the expression of *ENO2* was upregulated > 2-fold in HCT116-$PTEN^{-/-}$ cells relative to isogenic HCT116-$PTEN^{+/+}$ cells (Figure 6A and Supplementary Figure S10D). *ENO2* encodes γ-enolase, also known as phosphopyruvate dehydratase, and acts in the penultimate step of glycolysis. Enolase activity is essential and, in humans, is imparted by the functionally redundant genes *ENO1* and *ENO2*. Consistent with our observations in human cells, we also observed a striking negative genetic interaction between *Pten* and *Eno1* in $Pten^{-/-}$ mouse embryonic fibroblasts compared with $Pten^{+/+}$ isogenic mouse embryonic fibroblast cells (data not shown). To further explore the mechanism of this genetic interaction, we knocked down ENO2 protein expression with two independent shRNAs targeting *ENO2* message, and observed a significant reduction in phospho and total AKT levels, but only in $PTEN^{-/-}$ cells (Figure 6D and Supplementary Figures S10E–F). These results are consistent with a model, whereby ENO2 may act in a negative feedback loop to activate AKT and maintain a high rate of glycolysis (Supplementary Figure S10G).

We extended the same analytical approach to identify negative genetic interactions with oncogenic *KRAS*, and at

**Figure 5** Targetable vulnerabilities revealed through genetic interactions. (**A**) Pericentrin (red) and *NEDD1* (green) staining for centrosomes and DAPI staining (blue) to indicate nuclei in $PTTG1^{+/+}$ cells depleted after the knockdown of DHFR. (**B**) Representative images from a live-cell experiment, where $PTTG1^{+/+}$ cells were treated with 500 nM of MTX. Spindles were monitored with *TUBB2C*-GFP and chromosomes with *H2B*-BFP (see Supplementary Information for details). (**C**) Quantification of abnormal spindle morphology in $PTTG1^{-/-}$ cells expressing a negative control shRNA, shLACZ or $PTTG1^{+/+}$ cells expressing shLacZ, sh*DHFR* or treated with 500 nM MTX. At least 100 cells were analyzed for each experiment ($n = 3$), **$P < 0.01$, $\chi^2$-test. Scale bar, 6 µm. (**D**) Matrix of relative expression levels of indicated genes (columns) in isogenic cell lines (rows). Red indicates levels are up and blue levels are down in resistant versus sensitive cells. (**E**) Schematic depicting the effect of Cetuximab (CTX) inhibition on EGFR-RAS signaling. (**F**) Experimental workflow for the identification of synthetic lethal interactions with CTX in LIM1215 cells. (**G**) Scatter plot of zGARP scores for CTX-treated and -untreated LIM1215 cell lines from primary screens. Genes that are synthetic lethal with CTX ($P < 0.05$) are highlighted in red, whereas genes that are essential in both untreated and treated lines are highlighted in blue. (**H**) Synthetic lethal effect of CTX and CD83 shRNA in LIM1215 cells. Viability/cell number is depicted relative to the negative control shLacZ. Error bars represent 1 s.d. *$P < 0.05$ and **$P < 0.001$ calculated using a one-tailed Student's *t*-test ($n = 3$). (**I**) Synthetic lethal effect of CD83 knockdown on $KRAS^{G13D/-}$ cells is shown. Viability/cell number is depicted relative to the negative control shLacZ. Error bars represent 1 s.d. *$P < 0.05$ and **$P < 0.01$, one-tailed Student's *t*-test ($n = 3$). NS, not significant. (**J**) Rescue experiment showing the on-target effect of shRNAs targeting CD83, including sh1-*CD83* and sh2-*CD83*. $KRAS^{G13D/-}$ cells expressing GFP-tagged CD83 or CD-83-shR were counted by flow cytometry 6 days after infection with sh1-*CD83*, sh2-*CD83* or shLacZ as a negative control. (**K**) Overexpression of either GFP-tagged wild-type CD83, CD83-GFP or GFP-tagged shRNA-resistant CD83 (shR-CD83-GFP) protein in $KRAS^{G13D/-}$ cells showing correct membrane localization. (**L**) Experimental setup for validation of the negative genetic interaction between CTX and *CD83* in an *in vivo* xenograft model. (**M**) Ten mice were dissected for each group and tumors were weighed ($n = 20$). Error bars represent 1 s.d. (**N**) Schematic depicting possible genetic relationship between *EGFR-KRAS* and *CD83*.

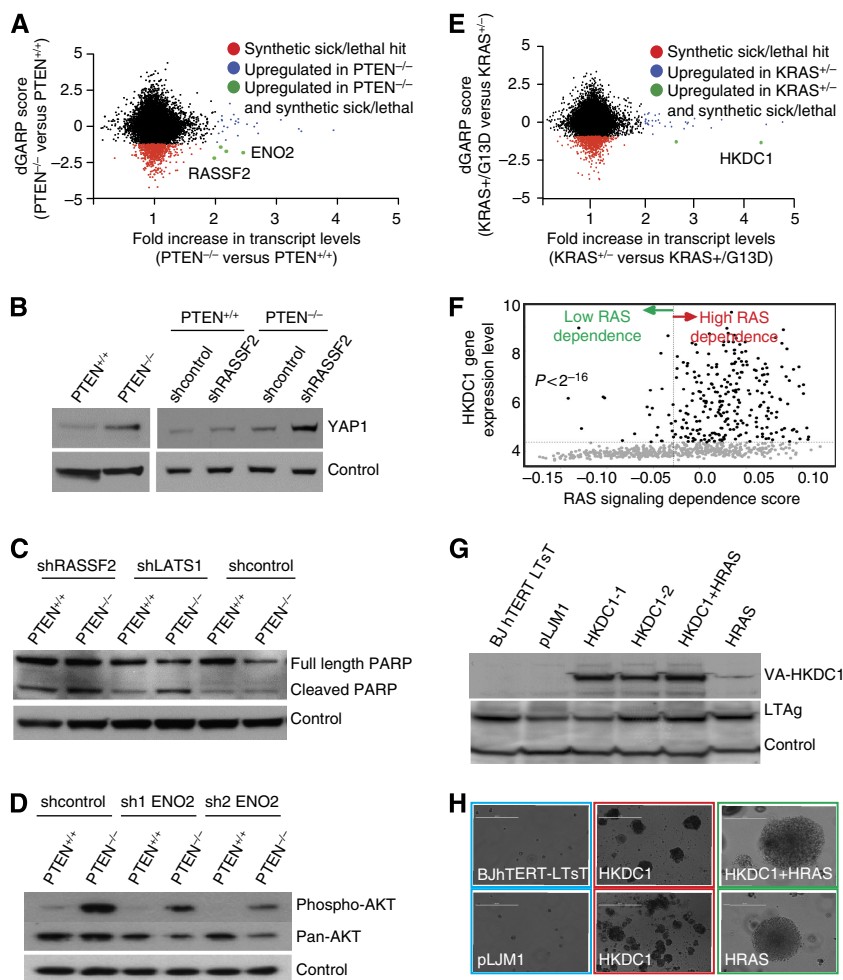

**Figure 6** Differential essentiality versus differential expression. (**A**) Scatter plot of fold-change in transcript levels in $PTEN^{-/-}$ cells compared with $PTEN^{+/+}$ cells (*x* axis) versus dGARP (*y* axis). The red points indicate genes that are SSL with the $PTEN^{-/-}$ genotype, the blue color indicates genes that are highly expressed in $PTEN^{-/-}$ cells versus $PTEN^{+/+}$ cells, and the green color indicates genes that are synthetic lethal and increased in expression in the mutant $PTEN^{-/-}$ cells. (**B**) Western blotting of *YAP1* protein levels in $PTEN^{+/+}$ and $PTEN^{-/-}$ cells infected with *sh*RNA against LacZ or RASSF2. (**C**) Western blot of *PARP* cleavage levels in $PTEN^{+/+}$ and $PTEN^{-/-}$ cell lines after cisplatin treatment in indicated knockdown conditions. (**D**) Western blot of phospho-*AKT* and pan-*AKT* levels in $PTEN^{+/+}$ and $PTEN^{-/-}$ cells infected with *sh*RNA against LacZ or ENO2. (**E**) Scatter plot of fold change in transcript levels in $KRAS^{+/-}$ compared with $KRAS^{G13D}$ lines against the differential essentiality score (dGARP score). The red color indicates genes that are synthetic lethal, the blue color indicates genes that are highly expressed in $KRAS^{+/-}$ cells and the green color indicates genes that are synthetic lethal and increased in expression in the $KRAS^{+/-}$ lines. (**F**) *RAS* signaling dependence score versus *HKDC1* expression in CCLE data sets. Vertical line separates high and low *RAS* signaling dependence; horizontal line separates *HKDC1* 'expressed' and 'expression uncertain'. Top-right quadrants contain overabundance of samples with clear *HKDC1* expression and high *RAS* dependence. $P < 2.2 \times 10^{-16}$ calculated using Fisher's exact test. (**G**) Western blot analysis showing the overexpression of HKDC1 tagged C-terminally with the VA affinity tag and the LargeT antigen in different backgrounds. (**H**) Colony-formation assay in soft agar for *HKDC1* overexpression. *GFP* (pLJM1) and $HRAS^{G12V}$ were used as negative and positive controls.

the same time genes that are differentially expressed in $KRAS^{+/-}$ compared with $KRAS^{+/G13D}$ cells. *HKDC1*, a gene that encodes an uncharacterized hexokinase domain-containing protein, was found to be differentially essential in $KRAS^{+/G13D}$ cells and underexpressed >4-fold in $KRAS^{+/G13D}$ versus $KRAS^{+/-}$ cells, suggesting that the overexpression of HKDC1 in $KRAS^{+/-}$ is compensating for loss of KRAS$^{G13D}$ expression in these cells (Figure 6E and Supplementary Figure S11A). The human genome encodes five hexokinases, but only *HKDC1* is generally essential across a compendium of cancer cell lines previously screened in our lab, many of which harbor activating mutations in KRAS (Supplementary Figures S11B and C; Marcotte *et al*, 2012). Importantly, our results are consistent with previous observations using the hexokinase inhibitor

3-bromopyruvate, which is highly toxic to HCT116, DLD1, VACO432 and RKO cells carrying *KRAS* oncogenic mutations compared with their isogenic *KRAS* wild-type controls (Yun *et al*, 2009).

As some level of *KRAS* expression is essential in HCT116 cells (Supplementary Figures S11D and E), we hypothesized that the $KRAS^{+/G13D}$ cells may overcome the loss of the $KRAS^{G13D}$ allele by upregulating *HKDC1* expression. Analysis of published expression profiles revealed that *HKDC1* expression was upregulated in 807 cell lines and 2158 tumors samples with a high *RAS* dependency score (Loboda *et al*, 2010; Figure 6F and Supplementary Figures S11F and G). On the basis of these observations, we further hypothesized that HKDC1 may have some transforming potential. To test this, we

performed a colony-growth assay in anchorage-independent conditions and compared the relative ability of *HKDC1* and *HRAS*$^{G12V}$ expression to transform human BJ fibroblasts in the presence of large T antigen, a standard cellular transformation assay. Overexpression of *HKDC1* in BJ fibroblasts lead to small-to-mid size colonies in soft agar that were visible to the human eye but did not exceed a certain size (Figures 6G and H). Importantly, the expression of both *HKDC1* and *HRAS*$^{G12V}$ was synergistic, resulting in larger colonies than *HRAS*$^{G12V}$ expression alone (Figures 6G and H). These results indicate that upregulation of *HKDC1* contributes to anchorage-independent growth, and its expression tracks with *RAS* dependency in both cell lines and tumors. Taken together, our data reveal specific examples of genes that are both differentially essential and differentially expressed.

## Integrating differential essentiality with variable cancer genotypes

Up to this point, our study has focused on the generation of a high-confidence DiE network and validation of specific negative genetic interactions that represent various network properties. We next asked the extent to which the digenic relationships derived from the HCT116 isogenic lines are generalizable to other non-isogenic lines of different lineages. As human cancer cell lines are genetically and epigenetically diverse and there are several large-scale screens that have defined gene essentiality in non-isogenic lines of different lineages (Luo *et al*, 2008; Silva *et al*, 2008; Barbie *et al*, 2009; Brough *et al*, 2011; Cheung *et al*, 2011), we tested whether different negative genetic interaction profiles derived from isogenic HCT116 cell lines could facilitate the classification of non-isogenic cancer cell lines into groups with known genomic features. To explore this idea, we generated negative genetic interaction profiles for each query (dGARP; *P*-value < 0.05) and then evaluated whether differentially essential genes were enriched for genes more essential for proliferation across a panel of screens in breast, pancreatic and ovarian cancer cell lines (GARP; *P* < 0.05) previously generated in our lab, using the same pooled shRNA library and standardized screening approach (Figure 7A; Marcotte *et al*, 2012). As a proof of concept we focused on PTEN, because the PTEN/PI3K/AKT pathway is frequently activated in breast cancer cell lines (Brough *et al*, 2011), which were highly represented in our compendium of non-isogenic cell line screens (Marcotte *et al*, 2012). Importantly, unlike the signature derived from the other mutant lines, the *PTEN*$^{-/-}$ signature derived from the HCT116 isogenic screens (Figure 7A and Supplementary Figure S13) was significantly enriched amongst the top 5% of genes that were more essential for proliferation in cell lines known to be PTEN/PI3K mutant compared with cell lines harboring wild-type *PTEN/PI3K* genes (Figures 7B and C; *P* < 0.0002). Interestingly, the *PTEN*$^{-/-}$ signature classified several pancreatic cancer cell lines as more dependent on the PTEN/PI3K/AKT signaling pathway for proliferation (Figure 7B), even though some of them are known to have wild-type *PIK3CA* and *PTEN* loci (Cheng *et al*, 1996; Halilovic *et al*, 2010). Closer examination of the genes within this *PTEN*$^{-/-}$ profile yield a set of candidates that link

diverse biological processes such as transcription, ubiquitination, metabolism and differentiation (Figure 7D and Supplementary Table S9). These results indicate that negative genetic interaction profiling in isogenic cancer cells reveals gene sets that can group cancer cells into functionally relevant classes.

## Perspective

Systematically defining genetic interactions is a powerful way to link genotype to phenotype, frequently implicating previously uncharacterized genes to specific pathways and complexes. In yeast, the position and connectivity of genes on a genetic interaction network is highly predictive of gene function (Costanzo *et al*, 2010). Our negative genetic interaction network (Figure 2 and Supplementary Figure S2A) with a small set of queries in isogenic cancer cells recapitulated some general properties of model organism genetic networks. First, it is highly complex with query genes showing dozens of interactions on average. Second, genes with related biological functions are connected by genetic interactions more often than expected by chance. For example, a large number of interactions were shared among *BLM*, *MUS81* and *PTEN* query genes, all of which have functions in the DNA damage and repair pathways (Supplementary Figure S12G; *P* < 10e − 10). In fact, genes showing negative genetic interactions with *MUS81* or *BLM* were also more likely than random to have annotated functions involving DNA damage and repair processes (Supplementary Figures S12E and F; *P* < 0.005), indicating that these screens identified functionally related genes. Third, and consistent with classic genetic relationships, our negative genetic interaction network revealed 'between pathway' genetic interactions (e.g., *PTEN-ENO2*, *PTEN-RASSF2*, *EGFR/KRAS-CD83*, *FANCD2/USP1/UAF1-MUS81* and *BLM-TOPBP1*) as well as 'within pathway' genetic interactions (e.g., *PTTG1-ESPL1*, *PTTG1-TTC31* and *PTTG1-ZC3H13*). This study unveils the incredible potential for mapping genetic relationships in human cancer cells using existing technologies.

A major objective of mapping negative genetic interactions in tumor cells is to identify chemical–genetic relationships that reveal possible therapeutic approaches. We highlight a few implications of the differential essentiality map with a possibility of therapeutic application based on the *EGFR/KRAS-CD83*, *PTEN-ENO2*, *PTTG1-DHFR* and *KRAS-HKDC1* negative genetic interactions. Given the clinical relevance of MTX resistance, we suggest that *PTTG1* and *ESPL1* could serve as biomarkers, as these interactions are conserved between both human HCT116 cancer cells and yeast cells.

The notion of generating a systematic map of genetic interactions in cancer cells represents one of the missing links in genotype-directed cancer therapy (Bernards, 2012). Conceptually, one can imagine using the methodology presented in the present study to develop a large network of genetic interactions using many query cell lines, or using double RNAi as has recently been described (Bassik *et al*, 2013; Roguev *et al*, 2013). In fact, the recent development of the CRISPR genome editing technology will be highly beneficial in constructing mutant isogenic lines for any gene of interest (Cong *et al*, 2013; Mali *et al*, 2013). In addition, chemical

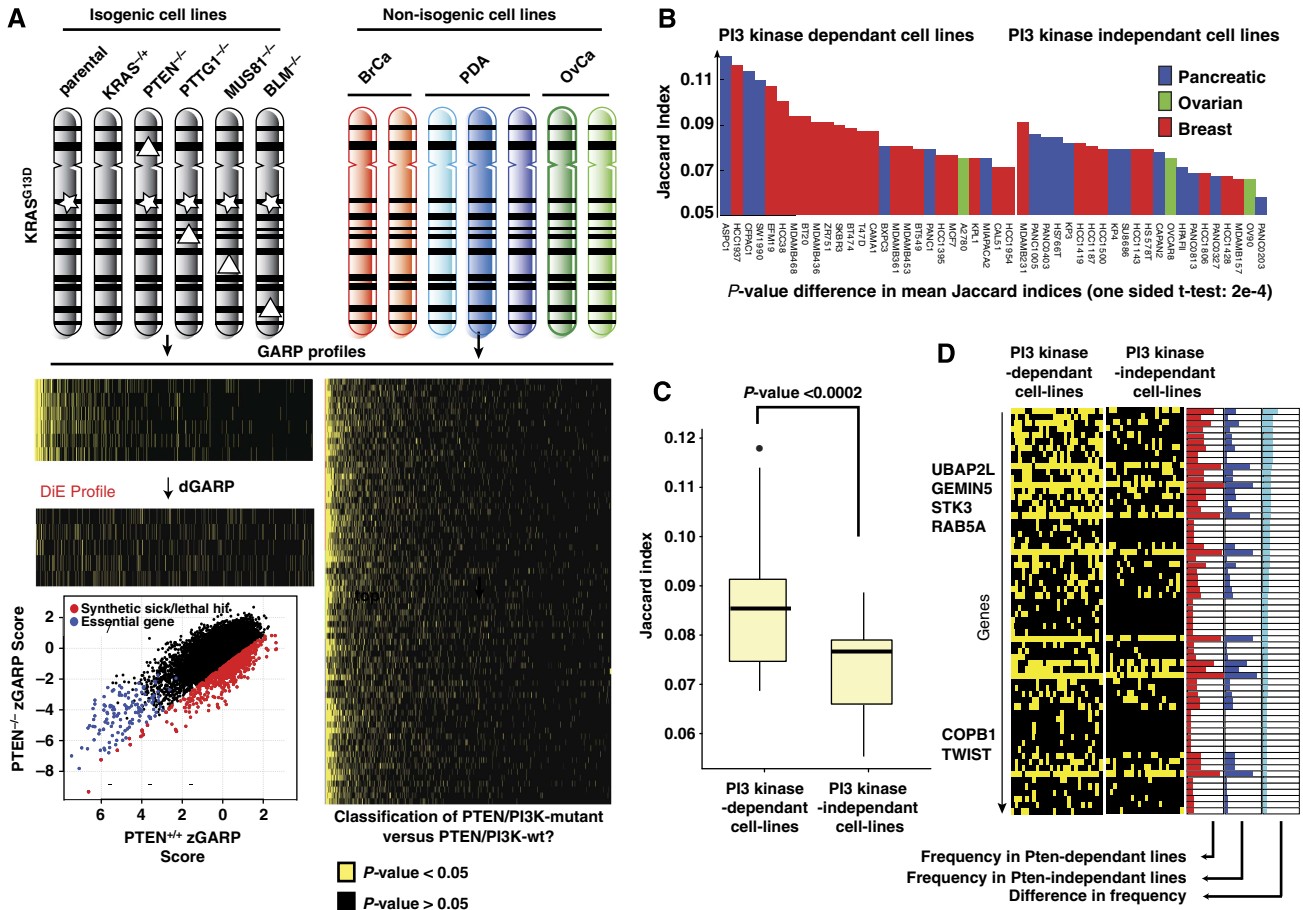

**Figure 7** Inferring genetic dependencies on variable cancer genotypes. (**A**) Schematic outline showing GARP profiles derived from isogenic cell lines on the left and essential gene profile from 72 non-isogenic cell lines on the right. GARP profiles for each cell line, both isogenic and non-isogenic, is represented as sorted dot plots with genes on the *x* axis and cell lines on the *y* axis. The yellow dots represent those genes that had a significant GARP score (*P*-value < 0.05). Genes are sorted by their frequency of being essential in the 72 cancer cell lines. The DiE profile contains the significant negative genetic interactions (*P*-value < 0.05, derived as pictured for the *PTEN*-null case in the lower left box: red dots, synthetic lethal or sick; blue, general essentials; black, not-influenced genes). We model that the enrichment of genes in the DiE profile from the isogenic cell lines will be a key feature in defining the genetic dependencies of the non-isogenic cancer cell lines. (**B**) Jaccard index (defined as the intersection between the DiE profile of *PTEN* and the essentials genes (*P* < 0.05) of non-isogenic lines divided by their union) exhibit a higher overlap on average for those lines that are dependent on PTEN/PI3K pathway (shift in mean is significant, *P*-value < 0.002). (**C**) The Jaccard index distribution for both the PTEN-dependent and PTEN-independent sets are plotted as box plot. Dots are extreme values (outliers; defined as having an Jaccard index more than 3/2 times higher as the upper quartile, which represents 75% of the data), the vertical lines represent the span of maximal to minimal value of non-outliers, the boxes span the upper (75% of the data) to lower (25% of the data) quartiles, the vertical line represents the median. (**D**) Heat map showing distributions of genes that exhibit negative genetic interaction with PTEN and are at the same time more frequently essential (*P* < 0.05) in the PTEN/PIK3-dependent cell lines compared with PTEN/PI3K-independent lines. Red color histogram shows the frequency in cell lines that are PTEN/PI3K dependent, blue represents the PTEN/PI3K-independent cell lines. Light blue bars give the difference in frequency between the two sets according to which the heat map is sorted (only genes with a difference in frequency > 0.1 and more frequent in PTEN/PIK3-dependent cell lines are shown).

genetic profiles can also be generated and overlaid on the map to understand the drug's mode of action, and provide pivotal insight into potential mechanisms of drug resistance. As observed for the budding yeast model system (Baryshnikova *et al*, 2010), elaboration of a genetic interaction map in cancer cells should reveal that the position and connectivity of genes on the map provide a precise prediction of gene function, resolving pathways and identifying their regulators and functional connections. Recently, it has been suggested that genetic interactions may underlie the mystery of 'missing heritability' of common traits that has haunted human genetics (Zuk *et al*, 2012). Decades of research have taught us that genetic interactions have a fundamental role in biology;

hence, this insight comes at an opportune time in cancer research, as advances in technology permit the systematic mapping of genetic interactions in tumor cells, which will lead to an intense understanding of pathways underlying different cancers and reveal new strategies for therapy and prevention.

## Materials and methods

A detailed description of all the methods can be found in the Supplementary Information. Unless indicated, all the cell lines were based on the HCT116 genotype. All Supplementary Tables can be accessed using one of the following links (http://kimLab1.ccbr.utoronto.ca/projects/cancer_essential/ or http://moffatlab.ccbr.utoronto.ca/resources.php).

## Supplementary information

## Acknowledgements

We thank the members of the Moffat laboratory for their input and discussions. We thank Bert Vogelstein, Kiyoshi Miyagawa and Todd Waldman for the cell lines, and Eric Campeau and Dr Stéphane Gobeil for plasmids. We thank Iain Wallace for curating the inhibitor collection from different resources, Michael Costanzo for providing updated yeast genetic interactions and Harrison Levy for technical help. The centrin antibody is a gift from Jeffrey Salisbury. RA is supported by an Ontario postdoctoral fellowship from the Ministry of Research and Innovation. JTFY holds a Banting and Best Doctoral Scholarship from the Canadian Institutes of Health Research (CIHR). This research was supported through funds from CIHR (JM), Canadian Foundation for Innovation (JM, CB and BA), the Ontario Ministry of Innovation GL2 program (CB, BA and JM) and the Canadian Institute for Advanced Research (JM, BA and CB). JM holds a Tier 2 Canada Research Chair in Functional Genomics of Cancer.

*Author contributions:* FJV, JM, BVA and CB conceived the idea. FJV performed all the primary screens. RA, FJV and PMK performed the analysis of genetic interaction networks, ortholog mapping and analyzed the conservation of genetic interaction across different species. FJV and FSV performed the secondary validation screens, and generated and analyzed the expression data for all the cell lines. KRB analyzed the primary screen data. JTFY generated Figure 3B. SL and FJV.. generated Figures 4F and G. JHMK, HT and FJV generated Figures 6B, C and D. PM generated Figures 6G and H. AS and GB analyzed the correlation of expression data for *HKDC1* and *DHFR*. AB, YW and COB performed the mouse xenograft experiments. YF, FSV, AS, WL, AM, PM, AE, AD, TK, LP, AF, JW, PMK and MB provided technical support and reagents. AB, YW, ZW, CAO and JM conceived, designed and performed experiments related to Cetuximab. FJV and JM performed or assisted with all the analyses, and wrote and edited the paper with input from all the authors.

## Conflict of interest

The authors declare that they have no conflict of interest.

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
