## [Review Process File · Molecular Systems Biology]

A Negative Genetic Interaction Map in Isogenic Cancer Cell Lines Reveals Cancer Cell Vulnerabilities

Franco J. Vizeacoumar, Roland Arnold, Frederick S. Vizeacoumar, Megha Chandrashekar, Alla Buzina, Jordan TF. Young, Julian HM Kwan, Azin Sayad, Patricia Mero, Steffen lawo, Hiromasa Tanaka, Kevin R. Brown, Anastasia Baryshnikova, Antony B. Mak, Yaroslav Fedyshyn, Yadong Wang, Glauber C. Brito, Dahlia Kasimer, Taras Makhnevych, Troy Ketela, Alessandro Datti, Mohan Babu, Andrew Emili, Laurence Pelletier, Jeffrey L. Wrana, Zev Wainberg, Philip M Kim, Robert Rottapel, Catherine A O'Brien, Brenda Andrews, Philip M. Kim, Charles Boone, Jason Moffat

Corresponding author: Jason Moffat, University of Toronto

Review timeline:

Submission date:	24 May 2013
Editorial Decision:	08 July 2013
Revision received:	23 July 2013
Editorial Decision:	23 August 2013
Revision received:	01 September 2013
Accepted:	03 September 2013

Editor: Maria Polychronidou / Thomas Lemberger

Transaction Report:

1st Editorial Decision

08 July 2013

Thank you again for submitting your work to Molecular Systems Biology. We have now heard back from the three referees who accepted to evaluate the study. As you will see, the referees find the topic of your study of potential interest and are overall supportive. However, they raise a series of concerns and make suggestions for modifications, which we would ask you to carefully address in a revision of the present work. Several of the reviewers' comments refer to the need to better explain and justify critical assumptions and parameters involved in the data analysis and to provide additional controls. Importantly, as reviewer #3 has pointed out, the raw and processed data should become available in the supplementary material.

REFeree REPORTS:

Reviewer #1 :

Cancer cells carry many mutations that are not present in the normal counterparts and these mutations lead to specific vulnerabilities which can be therapeutically exploited. In the present paper the authors used RNAi screens to identify such vulnerabilities in a set of 6 cell lines, the HCT116

cell line and 5 derived cell lines each deficient in one defined gene. Specific genetic vulnerabilities of the loss-of-function cell lines are interpreted as synthetic lethal/sick genetic interactions with the deficient gene. The authors test hits from the primary screens by competitive growth assay, and present detailed follow-up experiments on a number of selected synthetic-lethal gene pairs.

While much work remains to broadly map out such cancer -specific vulnerabilities, the approach and resulting data will likely be of interest to the readership of Molecular Systems Biology, and the MS should be suitable for publication once the following concerns are addressed:

Major points:

- As listed in Supplementary Table 5, the rate of validation for genetic interactions is low (<25%), even though the criterion used for validation is arguably quite lax. In the "high-confidence" genetic interaction map presented in Figure 2, only validated (confirmed) genetic interactions should be included. Interactions that failed to be validated (e.g., PSMB5/KRAS) should not be included based on conservation in other organisms, especially since the authors report that conservation of genetic interactions is generally low. Since a lax threshold is chosen for validation, and multiple hypotheses are tested, the false-discovery rate of the validation strategy should be estimated and discussed in the text.

- While the authors convincingly validate selected synthetic-lethal gene pairs, it is not clear how many true genetic interactions are missed in the map presented in Figure 2 (false-negative rate, and the related issue of specificity of the reported genetic interactions.)

For example, different proteasomal subunits are reported to have genetic interactions with only one query gene, but not the others.

If this result is accurate, it is potentially very interesting, since it may provide a basis for functional dissection of proteasomal subunit function.

Alternatively, thresholding of a noisy data set may have led to seemingly specific genetic interactions, where in reality each of the query genes has a negative genetic interaction with overall proteasome function.

The authors should address this issue ideally experimentally by, for example, conducting competition growth assays for a set of "specific" genetic interactors in all 5 query cell lines.

Minor point:

- The rationale for choosing the query cell lines, and their origin should be described in more detail in the main text - especially with respect to the KRAS mutant and deficient lines.

Reviewer #2:

This work presents the largest screen to date of negative genetic interactions in human cell lines. The approach taken is to transduce shRNA pools, comprising nearly the entire human genome of expressed genes and representing the 'prey' genes, against a set of five isogenic HCT116 cell lines, each homozygously mutated in one of five cancer-associated genes, PTTG1, BLM, MUS81, PTEN and KRAS, representing the 'bait' or 'query' genes. The 'real' way to test genetics interactions would be double shRNA knockdowns, but double shRNA knockdowns are not yet technically feasible, so the approach here of single shRNA knockdown in a suitably mutant cell line represents the cutting edge technology. The five isogenic cell lines that were screened may seem like a small number, but represents perhaps the largest to date genetic interaction screen in human cells.

Having then created a genetic interaction network around the five query genes, and having applied suitable and necessary control experiments to demonstrate that off-target effects were not predominant, as well as verifying a subset of the genetic interactions by more direct yet more cumbersome means, the result is a Differential Essentiality (DiE) network of genetic interactions (the name seems awkward). In what then follows the network is explored for examples of novel and unforeseen biological connections which are, in tour-de-force of validation, tested experimentally by multiple approaches to confirm the novel biology. For example, a conserved (in yeast) negative

genetic interaction between BLM and TOPBP1 was shown experimentally to cause accumulation of DNA damage. The expectation from the DiE network that the query gene MUS81 functions in removal of intrastrand cross-links was confirmed by multiple experiments. The number of examples is too many to list them all here. So chock full of biological validation is this work that in less circumspect hands might have resulted in two or three compelling papers in Cancer Research or Oncogene, not the single even more compelling paper under review here.

By the end the power of the genetic interaction network mapping approach undertaken here has been amply demonstrated. This work should be the first in a series of effective maps in genetic interactions pertinent to cancer.

While pretty much the DiE network has been optimally mined for biological insight, one intriguing observation that is never mentioned (Figure 2) is the profusion of 'housekeeping' RNA processing and ribosomal protein genes showing negative genetic interactions with the query genes, KRAS and MUS81 in particular. Do these interactions arise because of a general reduction in gene expression, with the query genes becoming harder hit than other cellular genes? Or is there something specific in these interactions, say a particular splice form of a query gene needed for activity not being produced? Or something else altogether? In their yeast genetic interaction maps was a similar enrichment for RNA processing genes noted? Obviously the questions cannot be answered in this work, but the enrichment is so striking that comment is called for.

Reviewer #3:

This is an interesting study that describes shRNA screens in 5 different isogenic cell lines carrying deletions of single cancer genes. Similar screens have been performed before for individual mutations or in complex genetic backgrounds, but the real strength here is the analysis of multiple screens all performed in the same isogenic cell line. There is extensive follow up validation of individual hits, and the study will overall stimulate further work in this very important field.

Minor comments and suggestions:

"The ultimate goal of developing a universal genetic interaction network for cancer cells will require a standardized approach that will serve to build a reference network of di-genic interactions in a common genetic background." I'm not sure this is the ultimate goal. Wouldn't it be better to identify interactions that reproduce across different genetic backgrounds (or to at least be able to predict the backgrounds in which an interaction will or will not have a large effect)?

"For each replicate screen, we examined multiple time points as the populations proliferated and evolved in culture and observed very good correlation between replicates (figs. S1A,B; R=0.9-0.99 for replicates)"
How does this compare to the correlations between different genotypes?

"Using stringent negative dGARP scores ($p < 0.05$) (figs. S1L-P and Table S2) and filtering for mRNA target gene expression (Table S3)"
I think the logic of this needs explaining in the main text. Why do the authors restrict themselves to genes differentially expressed between the parental and query cells - surely this will miss many genuine genetic interactions (e.g. this was not done in yeast). Is there any (systematic) empirical evidence that these interactions are indeed more reproducible?

"A total of 826 genetic interactions were tested across five assays and 200 unique negative genetic interactions (24.2%) were confirmed to differentially decrease the fitness of the query cells compared to the parental cells (80% CI; Figs 1M-Q, figs. S2H-L and Tables S4-S5)."
Why the choice of 80% CI? What's the number at 95% CI?

"Remarkably, clusters of densely connected functional modules emerged for each

of the five query genes (fig. S6A-E and Table S8)."
Delete 'remarkably'. (This is the expectation from model organism studies.)

Fig 7. I think here a specificity control is needed - i.e. if you take the hits from the KRAS, BLM etc isogenic cell line screens rather than from the PTEN isogenic screens, how well do they classify the PTEN/PI3K-mutant cell lines?

Importantly, the raw and processed (filtered, confirmed) data should be made available as supplementary tables to allow others to analyse the data/test data analysis methods/compare this data to their own screens.

1st Revision - authors' response

23 July 2013

Reviewer #1:

Cancer cells carry many mutations that are not present in the normal counterparts and these mutations lead to specific vulnerabilities which can be therapeutically exploited. In the present paper the authors used RNAi screens to identify such vulnerabilities in a set of 6 cell lines, the HCT116 cell line and 5 derived cell lines each deficient in one defined gene. Specific genetic vulnerabilities of the loss-of-function cell lines are interpreted as synthetic lethal/sick genetic interactions with the deficient gene. The authors test hits from the primary screens by competitive growth assay, and present detailed follow-up experiments on a number of selected synthetic-lethal gene pairs.

While much work remains to broadly map out such cancer -specific vulnerabilities, the approach and resulting data will likely be of interest to the readership of Molecular Systems Biology, and the MS should be suitable for publication once the following concerns are addressed:

We thank the Reviewer for his/her comments. Please see our specific responses to each point below.

Major points:

- As listed in Supplementary Table 5, the rate of validation for genetic interactions is low (<25%), even though the criterion used for validation is arguably quite lax. In the "high-confidence" genetic interaction map presented in Figure 2, only validated (confirmed) genetic interactions should be included. Interactions that failed to be validated (e.g., PSMB5/KRAS) should not be included based on conservation in other organisms, especially since the authors report that conservation of genetic interactions is generally low. Since a lax threshold is chosen for validation, and multiple hypotheses are tested, the false-discovery rate of the validation strategy should be estimated and discussed in the text.

As the hits nominated for follow-up validation were selected based on several criteria (Supplemental Figure 2B), it is not possible to provide an *estimated* FDR value for the genes declared "significant", as was done, for example, in Bassik et al 2013 (PMID: 23394947). However, the validation rates were included in Supplementary Table 5, and ranged from 14.4% to 46.2%. Accordingly, these validation rates provide an *observed* FDR of between 0.4 and 0.76, depending on the threshold selected in the validation assay (80% CI or 95% CI). The 80% CI corresponds to >2 standard deviations calculated from the mean of 25 negative controls.

Unlike an estimated FDR, our observed validation rate, and hence observed FDR, was obtained experimentally, which is on par with the validation strategy adopted by other groups. For example, using the same shRNA library both in their primary screen and in their secondary validation, Luo et al, (PMID: 19490893) observed a validation rate of 26%. In contrast, while our primary screen was based on the TRC shRNA library, validation was carried out with a completely independent siRNA reagent (Dharmacon) and still yielded an ~24% validation rate.

There are 3 classes of hits that we obtained from our primary and secondary screens (see Reviewer Table 1 below).

Reviewer Table 1

Interaction class	Primary Screen Hit	Secondary Screen Hit	Examples (screen in brackets)
Class 1	Yes	Yes	HKDC1 (KRAS ^{G13D/-}), CD83 (KRAS ^{G13D/-}), VCP (BLM ^{-/-}), KPNA6 (PTTG1 ^{-/-} and MUS81 ^{-/-})
Class 2	Yes	No	ESPL1 (PTTG1 ^{-/-}), TTC31 (PTTG1 ^{-/-}), ZC3H13 (PTTG1 ^{-/-})
Class 3	No	Yes	RASSF2 (KRAS ^{G13D/-}), SUPT5H (PTTG1 ^{-/-})

With respect to representing the conserved interactions in Figure 2, we strongly feel it is important to retain these interactions in the network as several examples that we explored in great detail were conserved but ended up as Class 2 genes. For example, the PTTG1-ESPL1, BLM-TOPBP1, PTTG1-DHFR, and PTTG1-TTC31 were all conserved. None of these interactions validated in the secondary screen, but they all validated in careful follow up experiments described in the main text. We clearly indicated in Supplemental Table 7 whether genes shown on the network in Figure 2 were confirmed, conserved, or both. We would argue that it is important to retain genes on the network presented in Figure 2 where there is evidence from model systems that these interactions are potentially conserved.

- While the authors convincingly validate selected synthetic-lethal gene pairs, it is not clear how many true genetic interactions are missed in the map presented in Figure 2 (false-negative rate, and the related issue of specificity of the reported genetic interactions.) For example, different proteasomal subunits are reported to have genetic interactions with only one query gene, but not the others. If this result is accurate, it is potentially very interesting, since it may provide a basis for functional dissection of proteasomal subunit function. Alternatively, thresholding of a noisy data set may have led to seemingly specific genetic interactions, where in reality each of the query genes has a negative genetic interaction with overall proteasome function. The authors should address this issue ideally experimentally by, for example, conducting competition growth assays for a set of "specific" genetic interactors in all 5 query cell lines.

Genetic interactions, by definition, are dynamic and context dependent, and thus may not be detectable in all experimental conditions. Furthermore, the paucity of known true positive genetic interactions involving human genes prevents an extensive examination of the false negative rate. In the current study, the criteria we applied in hit selection were geared towards providing the most comprehensive genetic interaction set possible. However, we fully acknowledge that interaction studies in other tissue types and in additional experimental conditions will be required to provide complete coverage of the human genetic interaction space. Further, as has been discussed in compiling protein-protein interaction networks (Braun et al, PMID: 19060903) and in the Discussion section of our manuscript, additional technologies such as haploid cell lines, temperature-sensitive mutants, more efficient methods for generating knock-out strains (e.g. TALENs, CrispRs, etc) will likely be required to reveal the complete compendium of human genetic interactions.

Our primary and secondary screens are de facto set up to detect differences in sensitivity to knockdown of different proteasomal subunits. It certainly remains a possibility that we are splitting

hairs with our data, particularly in light of the fact that proteasome function is absolutely essential for cell proliferation. Nevertheless, our data supports the statistical argument that some proteasomal subunits may be more extensively required for fitness in a genotype-driven manner.

Importantly, there is already precedent for genotype-dependent sensitivity to specific proteasome subunit depletion in cancer cells (Nijhawan et al; PMID:3429351). Moreover, groups have shown that cancer cells respond differentially to proteasome inhibitors, strongly supporting the idea that proteasome dependency is tightly coupled to genotype. In fact, multiple myeloma patients generally respond well to proteasome inhibitors, but there is a wide range of efficacy in this tumor type indicating a clear differential response based on genotype. We agree with the Reviewer that this result is potential interesting and would even go further and speculate that the proteasome is not the only essential complex where some subunits display differential essentiality. The fact that a few proteasomal components were found as genetic interactions (ie: have differential/context-specific essentiality) in the current study might be interesting to pursue in follow-up studies, and we thank the reviewer for pointing this out.

As the reviewer suggested, we tested the specificity of several interactions across all the different backgrounds and show that hits identified in a given screen were mostly specific to that screen. We have now included some examples of Class 1, 2 and 3 genes (see Reviewer Table 1) of specific interactions that were examined in multiple secondary screens in Supplementary Figure 2H-2K and include a paragraph in the main text on pages 9-10 (see yellow highlighted text).

Minor point:

- The rationale for choosing the query cell lines, and their origin should be described in more detail in the main text - especially with respect to the KRAS mutant and deficient lines.

We have now included a sentence in the main text on page 8 indicating our rationale for choosing these query genes.

Reviewer #2 :

This work presents the largest screen to date of negative genetic interactions in human cell lines. The approach taken is to transduce shRNA pools, comprising nearly the entire human genome of expressed genes and representing the 'prey' genes, against a set of five isogenic HCT116 cell lines, each homozygously mutated in one of five cancer-associated genes, PTTG1, BLM, MUS81, PTEN and KRAS, representing the 'bait' or 'query' genes. The 'real' way to test genetics interactions would be double shRNA knockdowns, but double shRNA knockdowns are not yet technically feasible, so the approach here of single shRNA knockdown in a suitably mutant cell line represents the cutting edge technology. The five isogenic cell lines that were screened may seem like a small number, but represents perhaps the largest to date genetic interaction screen in human cells.

*Having then created a genetic interaction network around the five query genes, and having applied suitable and necessary control experiments to demonstrate that off-target effects were not predominant, as well as verifying a subset of the genetic interactions by more direct yet more cumbersome means, the result is a Differential Essentiality (DiE) network of genetic interactions (the name seems awkward). In what then follows the network is explored for examples of novel and unforeseen biological connections which are, in tour-de-force of validation, tested experimentally by multiple approaches to confirm the novel biology. For example, a conserved (in yeast) negative genetic interaction between BLM and TOPBP1 was shown experimentally to cause accumulation of DNA damage. The expectation from the DiE network that the query gene MUS81 functions in removal of intrastrand cross-links was confirmed by multiple experiments. The number of examples is too many to list them all here. **So chock full of biological validation is this work that in less circumspect hands might have resulted in two or three compelling papers in Cancer Research or Oncogene, not the single even more compelling paper under review here.** By the end the power of the genetic interaction network mapping approach undertaken here has been amply demonstrated. **This work should be the first in a series of effective maps in genetic interactions pertinent to cancer.***

We appreciate the Reviewer's supportive comments.

While pretty much the DiE network has been optimally mined for biological insight, one intriguing observation that is never mentioned (Figure 2) is the profusion of 'housekeeping' RNA processing and ribosomal protein genes showing negative genetic interactions with the query genes, KRAS and MUS81 in particular. Do these interactions arise because of a general reduction in gene expression, with the query genes becoming harder hit than other cellular genes? Or is there something specific in these interactions, say a particular splice form of a query gene needed for activity not being produced? Or something else altogether? In their yeast genetic interaction maps was a similar enrichment for RNA processing genes noted? Obviously the questions cannot be answered in this work, but the enrichment is so striking that comment is called for.

The reviewer has highlighted a very interesting point, which has been a major challenge in interpreting genetic interactions. As with many such screens, the list of newly identified genes include those for which links to a given cellular process are not immediately clear, and those that reveal tantalizing new connections. The former group includes genes involved in metabolism and ribosome biogenesis, and although perhaps difficult to understand mechanistically, they underscore the complexity of the fully integrated cellular system. In fact, these house keeping genes do appear in the yeast interaction network as well (Costanzo et al, 2010; PMID: 20093466).

Reviewer #3:

This is an interesting study that describes shRNA screens in 5 different isogenic cell lines carrying deletions of single cancer genes. Similar screens have been performed before for individual mutations or in complex genetic backgrounds, but the real strength here is the analysis of multiple screens all performed in the same isogenic cell line. There is extensive follow up validation of individual hits, and the study will overall stimulate further work in this very important field.

We thank the Reviewer for his/her thoughtful comments and address each point below.

Minor comments and suggestions:

"The ultimate goal of developing a universal genetic interaction network for cancer cells will require a standardized approach that will serve to build a reference network of di-genic interactions in a common genetic background." I'm not sure this is the ultimate goal. Wouldn't it be better to identify interactions that reproduce across different genetic backgrounds (or to at least be able to predict the backgrounds in which an interaction will or will not have a large effect)?

We agree with this statement and have re-phrased our statement (highlighted in yellow on page 6) to more accurately convey the importance of having a reference network to develop predictive dependencies for different genetic backgrounds.

"For each replicate screen, we examined multiple time points as the populations proliferated and evolved in culture and observed very good correlation between replicates (figs. S1A,B; R=0.9-0.99 for replicates)" How does this compare to the correlations between different genotypes?

The correlations between different genotypes and their corresponding replicates were represented as a clustergram in Supplementary Figure 1A. In general, replicates cluster together for any given genotype at a specific timepoint. Correlation coefficients between genotypes tend to drop over longer incubation periods.

"Using stringent negative dGARP scores ($p < 0.05$) (figs. S1L-P and Table S2) and filtering for mRNA target gene expression (Table S3)" I think the logic of this needs explaining in the main text. Why do the authors restrict themselves to genes differentially expressed between the parental and query cells - surely this will miss many genuine genetic interactions (e.g. this was not done in yeast). Is there any (systematic) empirical evidence that these interactions are indeed more reproducible?

The reviewer is correct that this was not done in yeast. However, to define a set of therapeutically

valuable targets, we made a systematic effort to couple synthetic lethality with differential gene expression. We were also very interested in determining if there is a trend between differential expression and differential essentiality. Although no general trend was observed, the overexpression level may reflect a compensatory mechanism for impaired functioning of the machinery responsible for maintaining the integrity of genetic information. This possibility would provide potential targeting strategies for cancer treatment. Indeed we found this to be consistently true for multiple examples validated in PTEN and KRAS screens. With respect to the reproducibility, we show that hundreds of expression samples available in the CCLE database (<http://www.broadinstitute.org/ccle/home>), the expression of HKDC1 is consistently high in those cell lines that have a high RAS dependency (Fig 6F and Supplementary Figure 11F-G) strongly providing an empirical evidence that these interactions are indeed more reproducible.

"A total of 826 genetic interactions were tested across five assays and 200 unique negative genetic interactions (24.2%) were confirmed to differentially decrease the fitness of the query cells compared to the parental cells (80% CI; Figs 1M-Q, figs. S2H-L and Tables S4-S5)." Why the choice of 80% CI? What's the number at 95% CI?

We chose 80% CI because everything in the 80% CI was >2 standard deviations from the mean of 25 mock transfections (ie. negative controls). We present the 95%, 80%, and 68% CIs in Supplementary table S5.

"Remarkably, clusters of densely connected functional modules emerged for each of the five query genes (fig. S6A-E and Table S8)." Delete 'remarkably'. (This is the expectation from model organism studies.)

Done (see yellow highlighted text on page 12).

Fig 7. I think here a specificity control is needed - i.e. if you take the hits from the KRAS, BLM etc isogenic cell line screens rather than from the PTEN isogenic screens, how well do they classify the PTEN/PI3K-mutant cell lines?

We thank the reviewer for this suggestion and have completed these analyses and included the results in Supplementary Figure 13D and E. It can be seen that none of the other signatures (KRAS, BLM, MUS81, or PTTG1) can classify the PTEN/PI3K-dependent and independent cell lines to the same extent as the PTEN signature. A sentence in the main text was changed (highlighted in yellow on page 22) to emphasize this point.

Importantly, the raw and processed (filtered, confirmed) data should be made available as supplementary tables to allow others to analyse the data/test data analysis methods/compare this data to their own screens.

All of the raw data from the secondary validation screens has been included in Supplemental Table 5. The raw data for all the primary screens will be available through the COLT website upon publication as was previously indicated in the Supplemental Information.

2nd Editorial Decision

23 August 2013

Thank you again for submitting your work to Molecular Systems Biology. We have now heard back from the referee who agreed to evaluate your manuscript. As you will see below, the reviewer feels that the main concerns have been satisfactorily addressed, with the exception of one remaining point which we would like to ask you to address in a revision of the manuscript.

REFeree REPORTS:

Reviewer #3:

I am happy with all of the authors' responses except the one copied below:

"Using stringent negative dGARP scores ($p < 0.05$) (figs. S1L-P and Table S2) and filtering for mRNA target gene expression (Table S3)" I think the logic of this needs explaining in the main text. Why do the authors restrict themselves to genes differentially expressed between the parental and query cells - surely this will miss many genuine genetic interactions (e.g. this was not done in yeast). Is there any (systematic) empirical evidence that these interactions are indeed more reproducible?

The reviewer is correct that this was not done in yeast. However, to define a set of therapeutically valuable targets, we made a systematic effort to couple synthetic lethality with differential gene expression. We were also very interested in determining if there is a trend between differential expression and differential essentiality. Although no general trend was observed, the overexpression level may reflect a compensatory mechanism for impaired functioning of the machinery responsible for maintaining the integrity of genetic information. This possibility would provide potential targeting strategies for cancer treatment. Indeed we found this to be consistently true for multiple examples validated in PTEN and KRAS screens. With respect to the reproducibility, we show that hundreds of expression samples available in the CCLE database (<http://www.broadinstitute.org/ccle/home>), the expression of HKDC1 is consistently high in those cell lines that have a high RAS dependency (Fig 6F and Supplementary Figure 11F-G) strongly providing an empirical evidence that these interactions are indeed more reproducible. "

>>Given that there is no trend between differential expression and differential essentiality, I don't understand the choice of this filtering when deciding which interactions to present. I think it's fine to indicate which are differentially expressed, but a primary hit interaction with and without differential expression should, I think at the moment, be treated with equal confidence and so by equally presented in the data tables and figures of the manuscript. It is misleading to only focus on the differentially expressed subset at this stage.

2nd Revision - authors' response

01 September 2013

"Using stringent negative dGARP scores ($p < 0.05$) (figs. S1L-P and Table S2) and filtering for mRNA target gene expression (Table S3)" I think the logic of this needs explaining in the main text. Why do the authors restrict themselves to genes differentially expressed between the parental and query cells - surely this will miss many genuine genetic interactions (e.g. this was not done in yeast). Is there any (systematic) empirical evidence that these interactions are indeed more reproducible?

In order to improve our ability to discover true positive genetic interactions, we used expression as a filter to weed out potential false positives. In other words, if a transcript was undetectable or in the noise, we tagged that particular genetic interaction in Supplemental Table S4 in the "Expression" column as "Expression below threshold". As described in the Supplemental Methods (pages 5-6), we set our noise threshold based on "non-expressed" probesets (e.g. intron sequences, miRNAs, etc.), which technically should not be detected with our mRNA probe preparations. Another possible way that we could have deployed expression data to reduce false positives in our data would be to use presence/absence calls, for example.

The Reviewer is correct that this was not carried out in yeast (at least with the non-essential knockout strains) because knockouts, not knockdowns, were used to detect genetic interactions. Since RNAi depends on the presence of the target transcript, filtering for low abundance transcripts seemed a logical way to help further filter the true positive genetic interactions in our datasets. We have now added a sentence in the main text to further clarify why we performed this specific filtering step. The following sentence can be found at the top of page 8.

“To reduce the number of false positives inherent to RNAi screens, we also performed genome-scale microarray gene expression profiling experiments on parental HCT116 cells as well as all five query cell lines to measure target mRNA levels, and used these levels to determine a threshold for presence/absence (see Supplemental Information). Therefore, using stringent negative dGARP scores ($p < 0.05$) (figs. S1L-P and Table S2) and filtering for mRNA target gene expression (ie. presence/absence; see Table S3), we generated a network.....”

The idea of linking differential essentiality with differential expression is separate and distinct from using target mRNA expression levels to filter out potential false positives from our primary pooled shRNA screens. In our original and revised manuscripts, we described an entire section in the main text on this point (see page 19 under heading “(e) *Differential essentiality versus differential expression*”) and considered differential expression as a distinct filter for choosing candidates for our secondary colored competition assays and for further in-depth followup (see page 9). Although no general trend was observed, as was stated in the main text, there were some stark examples of where differential essentiality was linked to differential expression. Some of these were validated in-depth as described in the main text. Overall and as described in Supplemental Figure S2B, very few candidate negative genetic interactions were selected on the basis of differential gene expression (e.g. 5 genes for PTTG1, 3 genes for BLM).